# Hotspots and Hot Moments of Metal Mobilization: Dynamic Connectivity in Legacy Mine Waters

Anita Alexandra Sanchez[1,2]*, Maximilian P. Lau[1,2], Sean Adam[1,3], Sabrina Hedrich[1,4], & Conrad Jackisch[1,3]

[1]Interdisciplinary Environmental Research Centre, Technische Universität Bergakademie Freiberg, Freiberg, 09599, Germany

[2]Institute of Mineralogy, Technische Universität Bergakademie Freiberg, Freiberg, 09599, Germany

[3]Institute of Drilling Technology and Fluid Mining, Technische Universität Bergakademie Freiberg, Freiberg, 09599, Germany

[4]Institute of Biosciences, Technische Universität Bergakademie Freiberg, Freiberg, 09599, Germany

*Correspondence to*: Anita Alexandra Sanchez (Anita.Sanchez@mineral.tu-freiberg.de)

**Abstract.** Monitoring and treatment of contaminated mine water conventionally focuses on end-of-pipe assessment and remediation techniques, at the downstream outlet of mining sites after closure. Conversely, the initial stages of pollutant release and their pathways within abandoned mines have been largely overlooked. This study examines subsurface mining-affected anthropogenic structures and the dynamic hydrogeochemical loadings in terms of temporal increases in concentration and drainage pathways within them, revealing how variable subsurface flow activation impacts metal(loid) mobilization and opens novel direct mitigation options. We identified complex hydrological patterns through the mine (Reiche Zeche, Ore Mountains, Germany) in which percolation paths were dynamically connected to the drainage based on flow conditions. Using in-situ sensors, hydrogeochemical monitoring and stable water isotopes, we reveal a hydrodynamic regime in which episodic shifts in subsurface connectivity govern metal(loid) mobilization from localized storage zones, ultimately controlling solute export to surface waters. We use concentration–discharge (C–Q) relationships, the Pollution Load Index (PLI), and hydrological concepts to evaluate metal transport during the annual pattern of flow regimes. Our analyses of event-scale C-Q patterns reveal site- and element-specific shifts in flow path activation in a very short time. Low flow periods are often considered low risk for contaminant mobilization, yet contaminant hotspots within poorly connected hydrological zones can emerge during these times. The resulting high pollution potential and solute accumulation are governed by the sequence and timing of crossing or exceeding a connectivity or flow threshold, as described by fill-and-spill and lotic-lentic cycle concepts. Notably, Zn loads (in terms of flux) during low flow, pre-flush periods reached values up to six times higher than median values. Preceding the flushing events, geochemical and microbial-mediated metal leaching create the spatially distributed contaminant stock, remobilized during reconnection events. With a large proportion of heavy metal loads occurring during low flow and especially just before the high flow (flush) period, source-related, decentralized water treatment structures become much more feasible than end-of-pipe solutions that require higher throughput volumes and multi-element filtering. This work also highlights the need for event-sensitive monitoring and treatment strategy options that prioritize internal system behavior to mitigate pollution risk in abandoned mines and other caverned hydrological systems.

## 1 Introduction

Metal mining has left a pervasive global legacy of water contamination, particularly in river basins downstream of historic and active metal extraction zones (Macklin et al., 2023; Sergeant et al., 2022). Mine drainage affects more than 23 million people and thousands of kilometers of rivers globally, with risks that span decades to centuries after mine closure (Macklin et al., 2025). Despite regulatory progress, abandoned mine sites often lack monitoring and management, leaving communities and aquatic habitats vulnerable to pollution pulses triggered by hydrological events or anthropogenic disturbances. Standard monitoring under the European Water Framework Directive (WFD) and conventional water quality assessments (LAWA, 2003) typically rely on infrequent, low-resolution measurements providing only limited snapshots of hydrogeochemical processes (Resongles et al., 2015).

The consequences are particularly evident in regions with long mining histories, such as the Ore Mountains of Central Europe. Here, as in many former mining areas, legacy pollutants from underground workings pose environmental threats long after extraction has ceased (Huang et al., 2023; Liu et al., 2014) for example in the form of diffuse and point-source runoff of acidic waters bearing high concentrations of metals and sulfates (Bozau and Liessmann, 2017; Haferburg et al., 2022). While much attention has focused on surface water systems downstream former mining sites, the internal hydrogeochemical dynamics of underground mine workings remain poorly understood, especially in relation to episodic contaminant mobilization and non-conservative transport (Hudson et al., 2018; Datta et al., 2016). Addressing these blind spots is critical for understanding pollution behavior in mining-impacted systems and for designing effective remediation strategies.

Despite visible surface effects, the contaminant sources and pathways within abandoned underground mines remain largely obscured due to limited accessibility. Seeping waters infiltrate the mining system through complex pathways along underground waste rock deposits. While percolating or flowing through a fractured system of pools and pathways, waters dissolve and transport various elements. This suggests that pollution is not created continuously and diffuse but instead governed by discrete, intermittent and dynamically connective pathways. With hydrologic connectivity (Freeman et al., 2007) and intermittency (Fovet et al., 2021) known to impose specific characteristics on water-mediated transport and turnover in soil and other environments (Turnbull et al., 2018), the hydrological processes underlying contaminant mobilization and dispersal in abandoned mines may be better described using the tools and concepts of fill-and-spill (McDonnell et al., 2021) or lotic-lentic cycles (Schmadel et al., 2018).

In natural hydrologic systems, drainage connectivity, which controls water and solute transport, is shaped by catchment topography and becomes activated under specific hydro-meteorological conditions such as antecedent moisture, precipitation, infiltration, and subsurface flow through soil and fractured rock (Knapp et al., 2020; Li et al., 2017; Musolff et al., 2017; Lemenkova et al., 2021). In mining systems, infiltration water often enters deep storage zones where percolation is retarted. As a result, near-surface signals such as rainfall or snowmelt become lagged (delayed before appearing in discharge), low-pass filtered (short, high-frequency variations are dampened), and threshold-dependent (hydrological or geochemical responses only occur once storage limits are exceeded). Given the dispersed flow paths through subsurface waste rock deposits and other

anthropogenic preferential flow paths, various fill-and-spill pools overlay. To decipher the diffuse source pattern, the closer
the analysis can get to the individual sources and the higher the temporal resolution, the clearer should the mobilization pattern
be revealed.
Stable water isotopes ($\delta^2$H, $\delta^{18}$O) are useful tracers for identifying flow paths, water pool mixing, and water-rock interactions
(e.g., Sprenger et al., 2016; Spangenberger et al., 2007; Clark and Fritz, 2007; Kumar et al., 2024). Though widely used in
ecohydrology, isotopic tools remain underutilized in mine drainage studies. We suggest that they could provide a promising
means of tracing complex contaminant sources and transport processes (Ghomshei and Allen, 2000; Allen and Voormeji, 2002;
Hazen et al., 2002). Similar to surface catchments, mine systems experience episodic flushing during reactivation of subsurface
flow paths, when accumulated contaminants are rapidly mobilized following re-wetting periods. These short-lived first-flush
events produce sharp concentration peaks before dilution or source depletion occurs (Merritt and Power, 2022; Bryne et al.,

75  2012).

Concentration-discharge (C-Q) analysis provides a complementary approach to characterize such flow-phase-dependent
behavior and have seen broad application in watersheds (Shaw et al., 2020; Rose et al., 2018; Godsey et al., 2009; Knapp et
al., 2020; Musolff et al., 2015). Stable C-Q relationships indicate chemostatic conditions, often linked to proportional
weathering increases (Godsey et al., 2009; Li et al., 2017) and homogeneous solute distribution (Herndon et al., 2015), whereas
enrichment or dilution patterns (chemodynamic behavior) reveal heterogeneity in solute storage and mobilization (Herndon et
al., 2015). However, recent work emphasizes that while C-Q tools are widely used, they are often under-contextualized and
over-interpreted in isolation, and their diagnostic value depends heavily on integrating them with additional hydrological and
biogeochemical information (Knapp and Musolff, 2024). Despite this, C-Q tools and associated metrics (e.g., C-Q slope,
hysteresis indices, ratio of the coefficients of variation of concentration and discharge ($CV_c/CV_q$)) remains minimal in
underground systems, where episodic connectivity complicate their interpretation, representing a methodological gap this
study seeks to address.
Building on our previous study that identified strong spatial and temporal heterogeneity in contaminant release within the
Reiche Zeche mine (Sanchez et al., 2025), this work investigates how dynamic hydrological and geochemical processes
generate short-lived but critical contaminant release events. We focus on identifying hotspots, defined as spatial zones of
disproportionately high contaminant accumulation, and hot moments, defined as short time periods when mobilization rates
are markedly elevated due to transient changes in hydrological connectivity (McClain et al., 2003). The overarching research
question guiding this study is: How can dynamic contaminant mobilization within underground mine systems be effectively
monitored and translated into targeted, in-situ treatment strategies that move beyond conventional end-of-pipe approaches?
We hypothesize that alternating hydrological flow phases control dynamic connectivity and thus metal mobilization, with C-
Q patterns revealing the behavior of localized pools within the mine. To address our research question and hypothesis, our
specific objectives were: (1) to characterize the temporal evolution of flow regimes and their influence on metal(loid)
concentrations and loads, (2) to determine the geochemical signatures associated with localized storage and release zones

(hotspots) and episodic release events (hot moments), and (3) to evaluate how phase-dependent flow and C-Q relationships can inform adaptive, near-source mine water treatment strategies.

Therefore, we performed 42 underground sampling campaigns and utilized in-situ sensors across four distinct flow paths for over two years. At one site, we conducted high-resolution and high-frequency monitoring using an in-situ UV-Vis spectrometer to capture transient fluctuations. This multi-scale hydrogeochemical approach integrates complementary event-sensitive methods, extending surface-hydrological tools such as C-Q analyses, and fill-and-spill and hotspot/hot moment concepts to a subsurface mine drainage setting. Ultimately, this study contributes a transferable framework for diagnosing contaminant risks in legacy mine settings and supports the development of adaptive, near-source water treatment strategies.

## 2    Methods

### 2.1  Study site and sample collection

Situated in the Ore Mountains of Central Europe, the historic Reiche Zeche mine site, 50.928° N 13.357° E, is one of the many old mines whose runoff flows untreated into streams that feed the Elbe river, one of the largest rivers in Europe (LfULG, 2014). This site was active in extracting high-grade minerals, specifically silver ore, and processing mine waste rock up until 1969. The host rocks comprise mica schist and gneiss intersected by polymetallic sulfide-quartz-carbonate veins containing pyrite, sphalerite, galena, and chalcopyrite, with minor arsenopyrite, barite, and fluorite (Baacke, 2001; Tichomirowa et al., 2010). These sulfide-rich assemblages are key sources of acid generation and metal mobilization, while secondary Fe-(oxyhydr)oxides formed under drainage conditions contribute to local attenuation. Following mine decommission, the lower sections of the adit system, which extend down to 1300 meters, became inundated with water up to the level of the central adit "Rothschönberger Stolln", which is accessible at approximately 230 meters below the surface at the Reiche Zeche mine shaft (Zhiteneva et al., 2016; Mischo et al., 2021). The mine now represents a flooded, multi-level system with complex subsurface flow pathways. This hydrological complexity, dispersed flow above extraction levels and preferential flow through waste deposits, makes the site ideal for addressing our research questions.

This study focuses on a single slanted vertical extraction structure which spans over three levels before reaching the central drainage adit (Fig. 1). These levels include: Level 1 (located 103 meters below the surface); Level 2 (149 meters below the surface); and Level 3 (191 meters below the surface). A total of 26 sites were selected for ongoing sampling, but four specific sites will be in the focus. The four locations, sites 1, 2, 3A, and 3B, were selected due to the presence of continuous and ample amounts of flowing water in comparison to the rest of the locations which were not as great in volume of flowing water (see images of four locations in Fig. S6). In one-to-three-week intervals, we conducted 42 sample campaigns to all sites from February 3$^{rd}$, 2022 to May 31$^{st}$, 2024. All data are reported in the B2SHARE Data Repository (Sanchez et al., 2026).

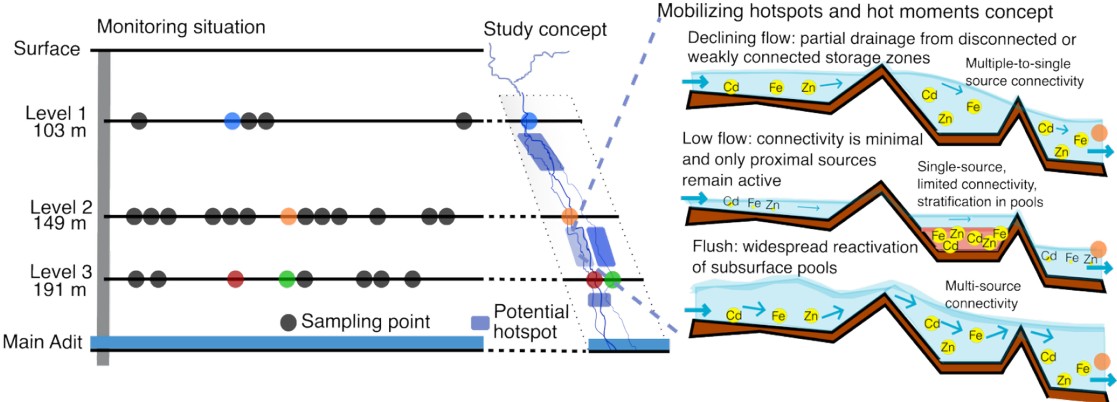


**Figure 1: Conceptual framework and study layout of the abandoned mine system levels above the main adit at Reiche Zeche. Left:**

**monitoring locations across three mine levels, with four sites (1, 2, 3A, 3B) selected for high-frequency sampling. Instrumentation at**

**all four sites included flow loggers, with additional high-frequency sensors and an autosampler deployed at site 2 for a 10-month**

**intensive monitoring period. Middle: concept of site placement along flow path through the mine and associated contaminant**

**hotspots (blueish purple zones). Potential hotspots along the flow paths to sites 3A and 3B are depicted in different color shades to**

**emphasize their distinct source zones, despite spatial proximity to site 2. Right: conceptual model of flow-phase-dependent fill-and-**

**spill connectivity. Blue arrows represent flow direction; shaded red areas indicate stratification. Flow regimes govern activation of**

**solute source zones, resulting in distinct connectivity patterns.**


## 2.2 Sampling design and conceptualization of site-specific dynamics

To unravel the internal dynamic hydrogeochemical characteristics of the abandoned ore mine and to interpret observed

heterogeneity in space and time, we focused on the four sites, enabling a vertical profile of hydrological connectivity within

the system (Fig. 1). Over two years of sampling, we measured discharge, isotopic composition ($\delta^2$H and $\delta^{18}$O), and dissolved

metal(loid) concentrations at all 26 locations (Fig. 1 and Table S1). These measurements form the basis for subsequent analyses

described later on. Initial observations revealed strong spatial heterogeneity in metal(loid) concentrations (Fig. S1) and

dynamic flow variability, which suggests the presence of transient contaminant hotspots and episodic connectivity. For a

process-based interpretation of these patterns, we developed a hotspot connectivity concept grounded in the fill-and-spill

paradigm (Fig. 1, right panel).

The Central European hydrological cycle (wet winters, dry summers) produces three recurring drainage phases in the mine:

flush (high flow), declining flow, and low flow. During low flow, hydrological disconnection allows solutes to accumulate in

lentic or weakly connected storage zones. Flush events re-establish connectivity, linking multiple pools and triggering

contaminant release (Sanchez et al., 2024), while declining flow reflects waning but still active transport. Stratification during

low and declining flows (Fig. 1) acts as a critical disconnection mechanism that can delay or abruptly initiate solute mobilization.

Although sites 3A and 3B are located on the same mine level, they receive water from distinct source zones shaped by geological structure, mining voids, and fracture networks. These differences lead to divergent C–Q dynamics and phases of the fill-and-spill cycle, reflecting contrasts in pool storage, reconnection timing, and redox conditions. Such site-specific variability underscores the need to analyze contaminant transport at multiple locations within the mine.

## 2.3 Hydrological data collection and analysis

To understand whether surface hydro-meteorological forcing translates into episodic contaminant release underground, we monitored both external conditions and internal mine discharge. Meteorological conditions are monitored in an automated station at the surface next to the central access shaft to the Reiche Zeche research and education mine. To avoid more complex hydrological modelling, a standardized water availability index, i.e. the Self-Calibrating Palmer Drought Severity Index (PDSI), was used to characterize the overall moisture conditions of the system and pre-event wetness levels (Wells et al., 2004; Palmer et al., 2016). The PDSI (adhered to as water availability index) values are determined by using reference potential evapotranspiration (FAO56 Penman-Monteith method) and precipitation data, and a simplified soil water balance model. This accounts for both short-term fluctuations and long-term storage effects with its self-calibrating structure allowing the effective storage capacity to adjust dynamically to the amplitude of the local weather variability. The magnitude of PDSI indicates the severity of the departure from normal conditions. A PDSI value greater than 1 represents wet conditions, while a PDSI value less than -1 represents dry conditions at the surface. The general dry and wet phases from the surface were compared with flow rate measurements from within the mine.

Continuous water level and flow monitoring was conducted at four sites within the Reiche Zeche mine using pressure sensors (Levelogger5, Solinst Georgetown) between February 2022 to March 2024. Sites 1 and 2 were equipped with plastic weirs for discharge measurement, while the existing carved spillways were used at sites 3A and 3B. Site-specific water level-discharge relationships were established (Henderson, 1966) and the resulting flow time series were smoothed using a Savitzky-Golay filter.

Additionally, to distinguish between recharge and stored drainage contributions, water stable isotopes ($\delta^2H$ and $\delta^{18}O$) were analyzed via cavity ring-down spectroscopy (L-2130i, Picarro Santa Clara) to trace water sources (See SI for details on discharge calculations and isotope methods). Comparison with the local meteoric water line (LMWL) and calculation of an offset from the general background concentration (centroid of all samples) and between the stations were used to assess seasonal recharge and drainage contributions and distinguish between precipitation-dominated and older subsurface waters.

## 2.4 Physico-chemical data collection and analysis

To evaluate contaminant concentrations and solute composition, we combined field parameters with laboratory analyses. Acid-washed HDPE bottles, pre-rinsed with deionized water, were used to collect water samples. For all samples, pH and

conductivity were measured (pH 340 and Cond 3310 sensors, WTW Weilheim). Prior to conducting analyses for dissolved organic carbon (DOC), dissolved inorganic carbon (DIC), and metal(loid)s, we filtered the samples using polyethersulfone filters with 0.45 μm pores (Filtropur S, Sarstedt Nümbrecht). The DIC and DOC concentrations were measured in triplicate for each sample using a total organic carbon analyzer (TOC-L series, Shimadzu Duisburg). For DOC measurements, we employed a high temperature combustion method, categorizing it as non-purgeable organic carbon (NPOC). This involved acidifying and then purging the samples with oxygen to expel inorganic carbon before the analysis. The precision of our data was validated by computing the standard deviation of the triple measurements, ensuring data reliability within the instrument's precision range (coefficient of variation < 2% and standard error < 0.1). We quantified metal(loid) concentrations using inductively coupled plasma optical emission spectroscopy (ICP-OES Optima 5300 DV Spectrometer, PerkinElmer Rodgau). For metal(loid) analysis, we prepared the samples with an addition of 1 mL of 2M nitric acid and included the following metal(loid)s in our analysis: iron (Fe), zinc (Zn), arsenic (As), copper (Cu), cadmium (Cd), lead (Pb), aluminum (Al), nickel (Ni), and manganese (Mn).These parameters allowed us to assess both geochemical conditions and contaminant levels under varying hydrological phases.

## 2.5 Automated sampling and high-resolution monitoring

To capture short-lived contaminant pulses that campaign sampling might miss, we complemented discrete sampling with automated high-frequency monitoring at site 2. An autosampler (6712 Full-Size portable sampler, ISCO Nebraska) was positioned at this site from May 16th, 2022 to February 14th, 2023 as an approach to avoid missing unseen aspects in the temporal dynamics of mine drainage water quality. The autosampler was calibrated to take a sample daily. 21 out of 24 1-L autosampler bottles were each filled with 10 mL of 2 M HCl prior to each start of the autosampler run to stabilize the metal(loid) solutions for measurements in the laboratory, while three autosampler bottles (one every seven days) were unacidified to record accurate pH and electrical conductivity measurements. The autosampler was filled every three weeks and 250 mL samples were collected from each bottle in the machine. Samples were filtered in the lab and prepared for further analyses. Prior to each new campaign, all 250 ml autosampler bottles were cleaned in a lab dishwasher and rinsed with deionized water. To complement this daily automated sampling, we submersed an online UV-Vis spectrometer probe (spectro::lyser V3, s::can GmbH Vienna; in the following simply termed spectrolyzer) in the flow channel from May 16th, 2022 to May 23rd, 2023 to record hourly absorbance measurements over a wavelength range of 200 to 720 nm at 2.5 nm increments.

To analyze and compare the spectral data obtained from the spectrolyzer with the metal(loid) concentration data collected by the autosampler at site 2 over time, we employed Quinlan's Cubist modeling (Kuhn and Johnson, 2013). Cubist, a rule-based method using spectrometric measurements, combines decision trees with linear models at the leaves, allowing for the prediction of continuous numerical variables. This approach was suited to our study because it handles non-linear relationships while maintaining interpretability. The modeling framework was applied to all analyzed metal(loid)s (see SI for details), while here we highlight cadmium as an illustrative example.

**2.6 Statistical and analytical framework**
**2.6.1 Hydrological phase classification**
To evaluate the influence of hydrological and geochemical drivers on contaminant mobilization, we divided the time series
into three hydrologically defined flow phases: low flow, flush, and declining flow. This classification was informed by
temporal patterns in discharge and water availability index values, observed consistently across the four monitoring sites.
Declining flow was characterized with the onset of dry conditions depicted by the water availability index turning negative.
Low flow marks the phase when the flow remains at very low rates although the surface system has started to recover from
the drying phase. Flush is defined by the onset of high discharge. The hydrological phases will be complemented with
geochemical phases later on.

**2.6.2 Pollution Load Index**
We further calculated the Pollution Load Index (PLI) to assess the cumulative level of metal(loid) contamination across the
four flow monitored locations. The PLI provides an aggregated measure of contamination by integration of the contamination
factors (CFs) of individual metal(loid)s, calculated as a ratio of observed metal(loid) concentrations to their respective
background reference values (Jahan and Strezov, 2018):
$$PLI = (CF1 \times CF2 \times CF3 \ldots\ldots CFn)^{\frac{1}{n}} \tag{1}$$
where CF= $C_{metal}/C_{background}$, and n is the number of metal(loid)s considered. A PLI > 1 indicates pollution, whereas PLI < 1
implies no contamination (Tomlinson et al., 1980). Reference concentrations (for all metal(loid)s except Al) were derived from
average values in the Elbe river at the Magdeburg station, located near the midpoint of the river, for the year 2022, obtained
from FGG Elbe Data Portal (Datenportal der FGG Elbe, 2025). This evaluation allowed us to assess relative contamination
levels at specific locations in the mine against a representative background from a major regional river.

**2.6.3 Concentration-Discharge analysis and indices**
Concentration-discharge (C-Q) relationships were analyzed in $\log_{10}$-$\log_{10}$ space to determine whether certain areas of the mine
disproportionately contribute specific metal(loid)s across the hydrological phases. The equation in $\log_{10}$-$\log_{10}$ form used to
describe general patterns between discharge and concentration magnitudes is as follows (Knapp et al., 2020):
$$\log_{10}(C) = \log_{10}(a) + b\log_{10}(Q) \tag{2}$$
with C as the concentration, Q as the discharge, and a and b as the intercept and slope values. The slope (b) value of each C-
Q relationship was used as the primary metric to evaluate site-specific solute behavior (Fig. 2).
Negative slopes (b<0) reflect source-limited dilution, as solute sources become insufficient at higher flows (Basu et al., 2010).
Positive slopes (b>0) reflect enrichment, pointing to transport-limited mobilization driven by large solute stores and increased
hydrological mobilization of solutes during increased hydrological connectivity (Pohle et al., 2021; Balerna et al., 2021). Flat
or near-zero slopes indicate chemostatic conditions, where concentrations vary little despite any changes in flow.

To further distinguish chemostatic from chemodynamic conditions, we calculated the ratio of the coefficients of variation of concentration and discharge ($CV_c/CV_q$). Following Musolff et al. (2015), chemostatic behavior is characterized by $-0.2 \leq b \leq 0.2$ and $CV_c/CV_q \leq 0.5$, whereas chemodynamic behavior corresponds to $-0.2 \leq b \leq 0.2$ and $CV_c/CV_q \geq 0.5$. Completely chemostatic conditions occur only when $b \approx 0$ and $CV_c/CV_q << 0.5$. While we adopt conventional thresholds as diagnostic guides, this term is used in our phase scheme to refer to segments whose behavior tends towards chemostatic-like signatures. To capture dynamic transport mechanisms, we additionally evaluated hysteresis in C-Q space using hysteresis index (HI) methods developed by Lloyd et al. (2016), Zuecco et al. (2016), and Roberts et al. 2023. Lloyd et al. (2016) was used as a directional index quantifying whether concentration responds earlier or later than discharge, while Zuecco et al. (2016) is an angle-based method that incorporates both discharge and magnitude of the loop, capturing asymmetry between rising and falling limbs. The HARP (Hysteresis Area, Residual, and Peaks) method from Roberts et al. (2023) provided a multi-component description of hysteresis area, lag symmetry, and peak timing, enabling a more holistic characterization of event-scale transport behavior. metrics for the hysteresis analysis. (Methodological details are included in the SI).

Hysteresis patterns reveal time lags between discharge and concentration, offering insights into hydraulic connectivity (Pohle et al., 2021) and mobilization processes at large (Lloyd et al., 2016). HI values typically range from -1 to +1, with positive values indicating clockwise hysteresis and negative values indicating counterclockwise hysteresis (Vaughan et al., 2017). Interpretation depends on the underlying C-Q behavior, whether concentrations rise or fall relative to discharge, and the hydrological context. Thus, hysteresis patterns were evaluated jointly with slope and $CV_c/CV_q$ ratios.

Together, these indices support systematic identification of shifts in contaminant sources, mobilization mechanisms, and hydrogeochemical memory, and form the basis for our specific C-Q conceptualization and geochemical phase classification.

### 2.6.4 Conceptualization of site-specific C-Q patterns

In order to interpret the event-scale hydrodynamics in our system, a conceptual basis is needed as flow response analyses alone do not fully capture the mechanistic processes that govern the supply of solutes to flowing waters. Observing C-Q dynamics through a connectivity-mediated lens (Fig. 2a) enables the identification of episodic transitions and characteristics that extend beyond general percolation and link to chemostatic and chemodynamic behaviors. Such transitions are best understood through the fill-and-spill concept in which water accumulates in isolated pools until a threshold is reached, after which overflow activates previously disconnected pathways (McDonnell et al., 2021). In our mine system, this behavior is further influenced by lentic-lotic cycling, where solute accumulation occurs during lotic channelized low flow conditions that maintain prolonged contact with metal-rich surfaces, followed by a transition to lentic, stratified pooling as water backs up behind internal thresholds. When these lentic layers spill, connectivity is abruptly re-established and stored solutes are rapidly flushed from the system, generating short-lived mobilization events (Schmadel et al., 2018). Depending on timing and degree of connectivity, these activation events may or may not coincide with elevated contaminant loads.

Based on a point-to-point analysis of the C-Q dynamics for each solute and site, we observed distinct time dependent
differences of how C-Q patterns evolved (Fig. 2b). Initial observations of hydrological flow and PLI patterns suggested four
recurring sequence-based behaviors: loading, flushing, dilution, and recession. Episodes of chemostatic-like behavior (varying
Q, stable C) also occurred, particularly during recession or periods of sustained connectivity. Here, our objective is not to
assign phases solely from abstract C-Q quadrant patterns, but to identify when in the event sequence these behaviors emerge,
and how they relate to underlying hydrological mechanisms such as threshold activation, lentic-lotic transitions, and fill-and-
spill cycles.

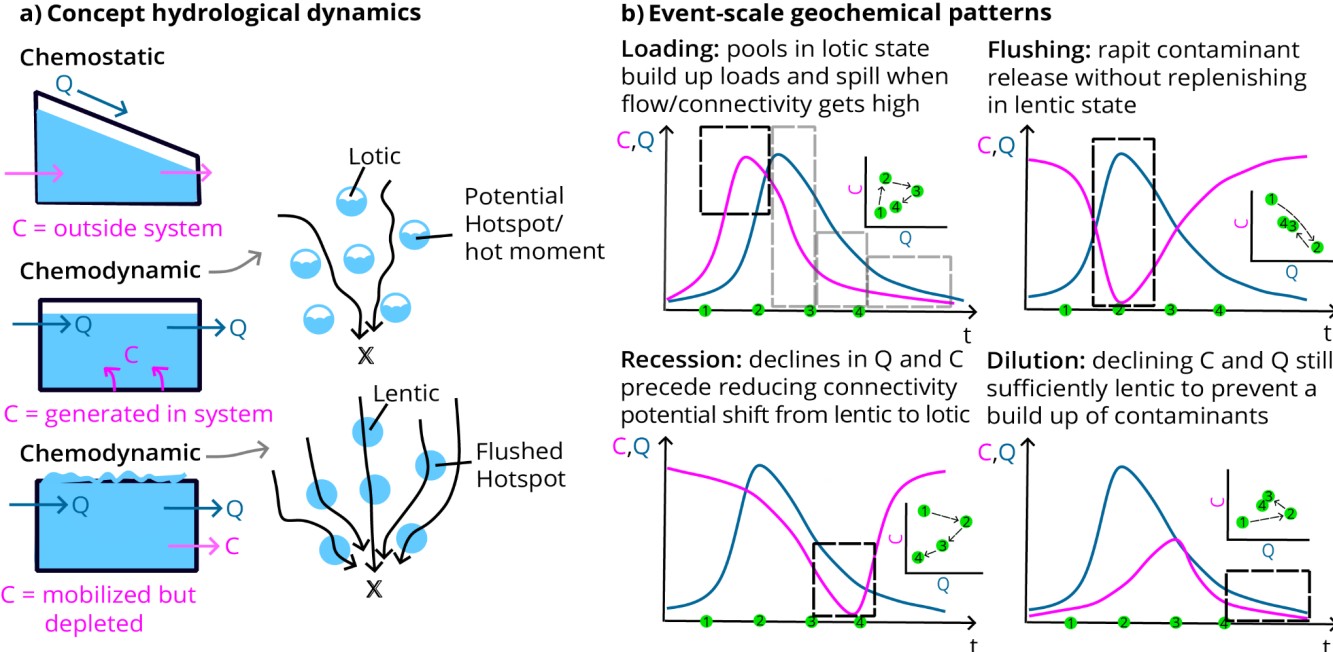


**Figure 2: (a) Conceptual representation of source and pool behavior with three dominant solute conditions. 1.** *External source input,*
*where* **concentrations (C) remain constant across a range of flows (Q) 2.** *Internal generation,* **where solutes accumulate within isolated**
**or weakly connected pools (e.g., lotic compartments) 3.** *Depleted pools,* **where previously enriched water masses are mobilized but**
**concentrations progressively decline as storage is exhausted. (b) Event–scale co-evolution of concentration and discharge. Time-**
**series patterns show loading, flushing, dilution, and recession patterns emerge during an event. Dark dotted box regions highlight**
**time windows in which the certain behavior is observed. Light dotted box regions in the first C-Q plot are shown to acknowledge**
**that other behaviors may also be present throughout different time points. Consecutive observations (green points) illustrate how**
**these patterns evolve through a hydrological event, and these transitions reflect the interplay of fill-and-spill activation, lentic-lotic**
**switching, and the spatial distribution of internal solute reservoirs. Together, these panels illustrate how event-driven changes in**
**connectivity and solute availability produce contaminant export behavior, including short-lived hotspots and hot moments.**

Applying traditional C-Q relationships in this temporal point-wise manner allows us to reveal how phase transitions structure contaminant export and generate hotspots of locally intensified metal release (Vidon et al., 2010) and hot moments of intensified metal discharge. These event- and site-specific patterns provided a process-based understanding of solute mobilization pathways that complemented the broader hydrological regime which aid in the development of our geochemical phase framework.

**2.6.5 Geochemical phase classification**

In addition to the hydrologically defined phases, we introduced geochemically defined phases to resolve finer-scale temporal variability in contaminant mobilization. To classify these phases, we examined time series trends in water availability index, discharge, PLI, C-Q slope, $CV_c/CV_q$ ratio, and HI behavior using the methods from Lloyd et al. (2016), Zuecco et al. (2016), and Roberts et al. (2023) for each site. This multi-metric approach follows recent guidance by Knapp and Musolff (2024), who emphasize that C-Q tools should not be interpreted in isolation but instead integrated with hydrological and geochemical context to avoid overgeneralization. By manually evaluating these parameters together over time and developing an automated classification algorithm, we identified characteristic patterns that delineate transitions between geochemical phases (Fig. S4). To quantify the instantaneous C-Q behavior, each pair of consecutive observations were evaluated using a point-to-point approach. For each segment, we calculated the C-Q slope and $CV_c/CV_q$ ratio using a five-point rolling window, as well as the hysteresis index values calculated on the time window surrounding each segment. These metrics were integrated into a hierarchical rule-based classification algorithm in which each segment was assigned a confidence score (0-1) based on how strongly its C-Q slope, $CV_c/CV_q$ ratio, and hysteresis behavior matched characteristic patterns for each phase. Phases were evaluated in priority order (flushing, loading, chemostatic, dilution, recession, and variable), with the first phase whose rules triggered being selected as the dominant phase of that segment.

These phases were interpreted within the chemostatic-chemodynamic framework of Musolff et al. (2015), in conjunction with the hydrological phase classification and the connectivity-based conceptual model (Fig. 2), ensuring that our phase classifications aid in process-based interpretations of contaminant transport and mobility. From these combined trends, we developed a working hypothesis in which these recurring geochemical phases emerge (Fig. S4):

1. Loading phase: Segments with increasing PLI values, negative C-Q slopes, and negative hysteresis during which flow is at its low or increasing were classified as loading. These conditions reflect moments where water resides long enough in isolated pools or channelized pathways for solute stores to accumulate. These segments correspond to the filling state before threshold activation.

2. Flushing phase: Segments with initially high PLI values which lower as discharge increases, positive C-Q slopes, and positive hysteresis were classified as flushing. These are short time windows where solute-rich lentic layers spill and mobilize accumulated solutes as connectivity rapidly expands. This aligns with threshold exceedance and the activation of previously disconnected domains.

3. Dilution phase: Segments with variably high flows and declining PLI values, and relatively high $CV_c/CV_q$ ratios with positive hysteresis were classified as dilution. Here, solute concentrations decrease due to mixing depleted lentic waters and less solute-rich flow. Connectivity persists but source reservoirs become progressively exhausted.

4. Recession phase: Segments with lowering flow and stable or slightly declining PLI trends, very low $CV_c/CV_q$ ratios, and low water availability index values were classified as recession. These segments typically occurred during periods of declining flow when connectivity contracts and solute exchange with source zones is limited.

5. Chemostatic: Periods where flow slightly varied but PLI, C-Q slope, and $CV_c/CV_q$ ratios remained relatively stable with low $CV_c/CV_q$ ($< 1$) and flat C-Q slopes, and low hysteresis indices were identified as chemostatic. These episodes occurred during sustained connectivity when reactive surfaces remain buffered and concentrations change minimally.

6. Variable sources: Segments that did not match the characteristic patterns of other phases, typically showing relatively stable flow and PLI trends with mixed or ambiguous changes in C-Q metrics were classified as variable sources. These segments indicated solute dynamics driven by processes other than flow magnitude alone.

Finally, points of maximum pollution potential were further identified where high concentrations were reached before transitioning to a substantial dilution behavior. These points represent the likely onset of a hot moment in contaminant mobilization. By integrating hydrological flow and PLI trends with point-wise C-Q evolution, fill-and-spill behavior, and lentic-lotic transitions, this phase classification and framework provides a mechanistic basis for interpreting episodic solute mobilization in underground mine systems.

**2.6.6 Data visualization**

All figures and data visualizations were produced using Python (v3.12), primarily with the pandas (McKinney, 2010; pandas development team, 2020) and plotly (Plotly Technologies Inc., 2015) libraries, and are reproducible with the code in the dataset (see Sanchez et al., 2026) and the code for the geochemical phase classification (Jackisch and Sanchez, 2026). The conceptual frameworks outlined in Fig. 1 and 2 further guided our analysis, motivating the structure of the results to follow the dynamics of flow-phase-dependent connectivity and site-specific contaminant mobilization.

**3    Results and discussion**

**3.1  Spatial and temporal patterns in hydrological and geochemical parameters**

The hydrological regime of the Reiche Zeche mine system exhibits pronounced temporal and spatial heterogeneity, driven by internal storage thresholds and episodic connectivity. Figure 3 summarizes key surface and subsurface hydrologic indicators, including the water availability index, precipitation, and discharge at the four monitored sites. Based on hydro-meteorological observations, declining flow, low flow, and flush phases of the mine drainage dynamics were identified. The declining phase is informed by the overall moisture regime starting with the landscape shift from wet to dry states and ending when both flow and dryness reach their minimum. These patterns were observed for two annual cycles (i.e, in 2022 and 2023). Notably, flush phases occurred in February 2023 – May 2023 and December 2023 – February 2024, marked by sharp increases in discharge

across all four sites. Discharge trends did not align tightly with precipitation inputs, suggesting delayed and non-linear hydrological responses (Milly et al., 2002; Bales et al., 2018). Direct reactions to surface storm events are very rare (Burnt et al., 2025) such that, although surface conditions transitioned from drought to wetter periods in early autumn, increased mine water discharge only became evident months later.

This temporal disconnect may reflect both delayed percolation to deeper layers and threshold-based fill-and-spill dynamics within vertically structured storage zones. The more pronounced discharge peaks observed at deeper sites (i.e., sites 3A and 3B) compared to shallower sites closer to the surface (i.e., site 1; Li et al., 2022) suggest that connectivity is not continuously active, but rather modulated by threshold exceedance, consistent with a fill-and-spill mechanism. The materials within these contaminant stores are easily entrained once hydrologic thresholds are crossed (e.g., rising water tables or shear stress increases), leading to a sharp but short-lived release pulse (Resongles et al., 2015). This illustrates how flow-phase-dependent changes in hydrological connectivity control source zone activation. Similar to braiding rivers, we expect parts of the system as being always drained and an increasing number of adjacent pools becoming connected with increasing water flow (Wilson et al., 2024). Permanently spilled sections have rather low metal(loid) concentrations, while temporarily disconnected sections act as niches for microbially mediated solving and hence elevated metal(loid) concentrations (Sanchez et al., 2025).

To further understand these patterns, we assessed spatial and temporal trends in the pollution load index (PLI) (Jahan and Strezov, 2018), which integrates multiple dissolved metal(loid) concentrations into a single risk metric (Fig. 3f). The PLI time series reveal clear site- and flow phase dependent variability. Notably, site 3A consistently exhibited the highest PLI values, often exceeding a value of 500, well above the pollution threshold of one (Tomlinson et al., 1980). These elevated PLI values declined sharply early in the flush events, consistent with dilution by low ionic strength water and enhanced mixing (Cánovas et al., 2007), and further suggesting solute buildup during low connectivity followed by rapid export when flow paths are reactivated.

We further classified the geochemical response at each site into distinct phases reflecting shifts in source zone activation and storage-release dynamics. These phases, derived from C-Q relationships and aligned with hydrologically defined flow conditions, include loading (characterized by temporal increases in solute concentration during low flow), flushing (rapid contaminant release upon reactivation), and dilution (declining concentrations with rising discharge). Additional phases include recession (post-flush declines in both flow and solute levels), chemostatic behavior (varying discharge with relatively stable concentrations), and variable phases (mixed or unstable transport conditions). This phase-based framework, illustrated by using PLI trajectories (Fig. 3e-f), offers a dynamic perspective on how internal thresholds and subsurface connectivity shifts modulate contaminant export, which is not in phase with the discharge dynamics. This approach challenges traditional drainage-based hypotheses by revealing that solute export is not a continuous seepage process, but rather a sequence of non-linear mobilization events tied to internal storage activation.

Site-specific patterns highlight important contrasts in system behavior. While site 1 (located on level 1 at 103 m below surface in our underground mine system) exhibited muted responses with the lowest PLI values, suggesting this point to depict the water entry into the subsurface deposit structures, sites 2, 3A, and 3B exhibited stronger temporal variability, indicative of

reactive source zones. In the lead-up to flushing events, elevated dissolved metal(loid) concentrations suggest slow leaching or desorption during storage-dominated phases (Pohle et al., 2021; Speir et al., 2024), while post-flush declines point to transient depletion of pools. $CV_c/CV_q$ ratios (Fig. 3g) reveal mostly chemodynamic behavior at sites 2, 3A, and 3B, with sharp peaks preceding flushing events and indicating unstable solute supply as connectivity expands. Correspondingly, large HARP hysteresis areas at these sites (Fig. 3h) reflect repeated short-lived mismatches between concentration and discharge, reflecting rapid shifts between lentic and lotic states. These sharp fluctuations in PLI and C-Q metrics and synchronized concentration responses (Fig. S1) highlight the episodic nature of contaminant release and support the view that reconnecting flow paths mobilize previously isolated geochemical reservoirs.

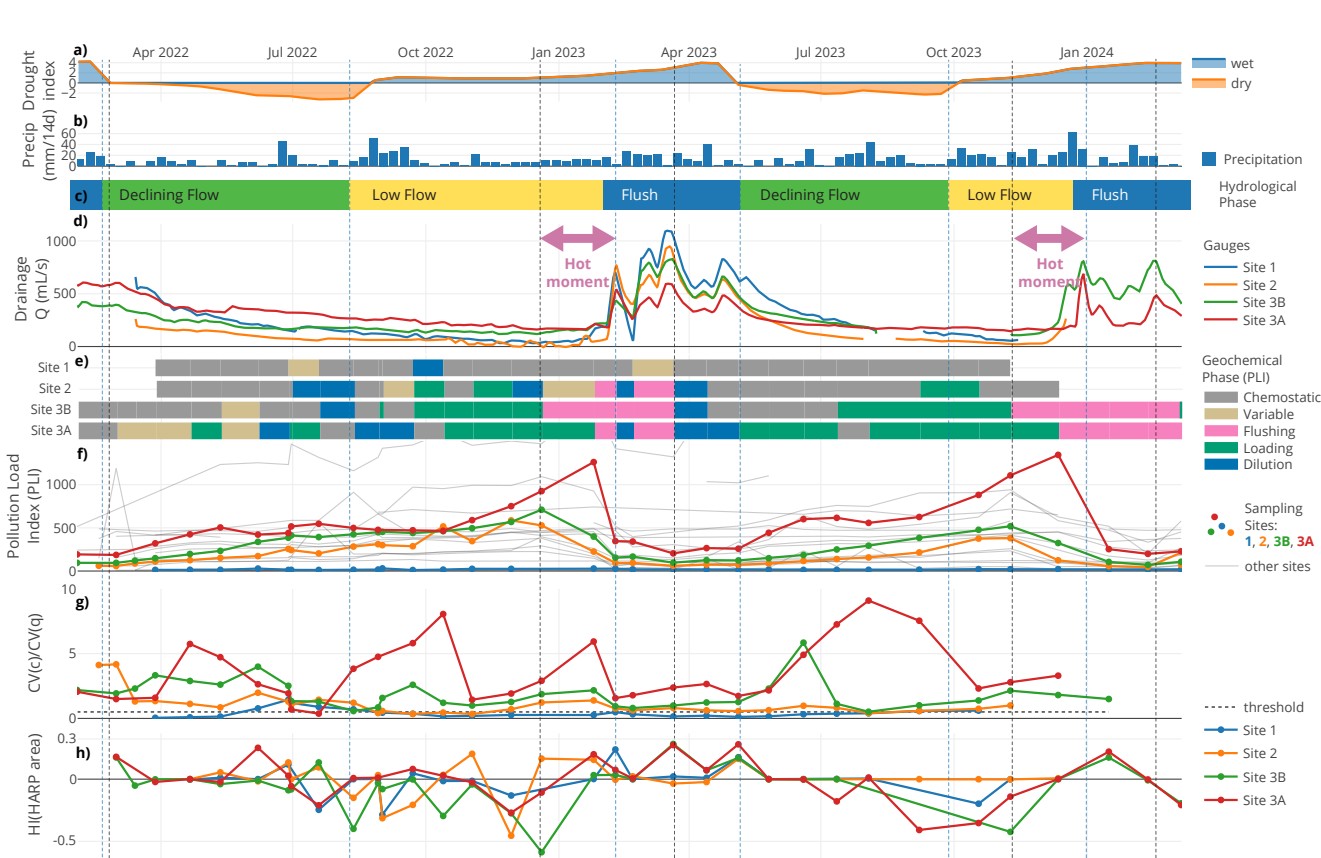

**Figure 3: Water and metal transport regime in the Reiche Zeche mine. (a) Dryness index as an indicator of the general water situation on the surface. (b) Weekly precipitation collected from Reiche Zeche, Freiberg weather station. (c) Hydrologically defined flow phases. (d) Discharge at the four sites in the Reiche Zeche mine. Pink arrows represent time periods when hot moments occur. (e) Geochemical hysteresis phases of pollution load index (PLI) at the four sites. Recession phase is not shown since it was not identified in the analysis. (f) PLI dynamics, (g) $CV_c/CV_q$ ratios (with a threshold of 0.5), and (h) Hysteresis Area from the HARP**

method are shown across the mine system. Individual sites are connected by black lines and colored lines are monitored flow sites
for sites 1, 2, 3A, and 3B.

### 3.2 Trends in metal load patterns and isotopic deviations

While PLI values present a general overview of metal(loid) behavior, looking into metal specific load dynamics (Fig. S3)
revealed key insights into the mechanisms controlling contaminant release in legacy mine systems. The daily Zn load (in terms
of flux, i.e. concentration times discharge) patterns observed at sites 3A and 3B (Fig. 4) exhibit sharp Zn load peaks shortly
before major flushing events, despite relatively stable or declining discharge conditions. While these pre-flush peaks occurred
during hydrologically low flow phases at both sites, the sites differ in terms of the contribution of each geochemical phase
within their fill-and-spill dynamics, such that this is more pronounced for site 3A than for site 3B. At site 3B, Zn dynamics are
less consistent with threshold-driven mobilization, suggesting that additional processes, such as dilution by younger recharge
waters or stratified storage, may have played a role as well. In contrast, site 3A shows sharper peaks that are better explained
by threshold-exceedance behavior. Once Zn accumulation is at its peak and mobilization begins, Zn loads drop abruptly,
reflecting rapid dilution or depletion of accumulated pools. This suggests that site 3A may be a more important target for water
remediation.
Quantitatively, these short 2-3 month intervals account for 50-56% of the total annual Zn load at site 3A and 34-35% at site
3B, despite occupying less than 25% of the time period. This highlights a strong fill-and-spill style signature, where
contaminants accumulate gradually under low connectivity and are then exported in intense but brief mobilization events.
Importantly, the overlay of flow and geochemical phases emphasizes how metal mobilization is driven by the timing and
sequence of hydrological reconnection.
To assess whether these mobilization pulses reflect deeper subsurface activation, we examined deviations in stable water
isotope compositions ($\partial$2H and $\partial$18O) between deeper sites (3A and 3B) and the shallower reference site 2 (Fig. 4c) as a
measure of hydrologic connectivity. Individual samples plotted similarly close to the overall mine water background and the
LMWL (Fig. S2), indicating only minor shifts in water sources and weathering interaction at this scale. However, large isotopic
differences, predominantly during low flow periods in 2022, support the hypothesis of weak connectivity and more isolated
subsurface storage compartments. These differences narrowed considerably during flush phases, indicating reactivation of
previously disconnected zones. Shorter distances among standard deviations further point to site 3A (or 3B) being influenced
by similar water sources as site 2 during flushing phases. These patterns extend prior applications of isotopic tracers beyond
surface water systems (Spangenberg et al., 2007; Hazen et al., 2002), demonstrating their value for characterizing episodic
hydrological activation and subsurface connectivity in mining environments.
These findings challenge the conventional focus on high-flow conditions as the primary drivers of contaminant export from
mining-impacted systems. While prior studies have highlighted the role of high flow in resuspending contaminated sediments
or altering water chemistry (e.g., via pH or redox shifts) (Hudson-Edwards et al., 1997; Dawson and Macklin, 1998), our
results point to a dominant role of low flow inputs from subsurface or groundwater sources. Unlike classical baseflow, typically

low in flow and constant in concentration, these low flow periods exhibited highly variable metal levels, indicating disproportionate contributions to contaminant loads from subsurface pools or intermittently connected sources. Similar conclusions have been drawn in other abandoned mine systems (Bryne et al., 2020), where metal fluxes were sustained or even amplified under low flow regimes, underscoring the need to reconsider assumptions about contaminant risk during non-flushing conditions.

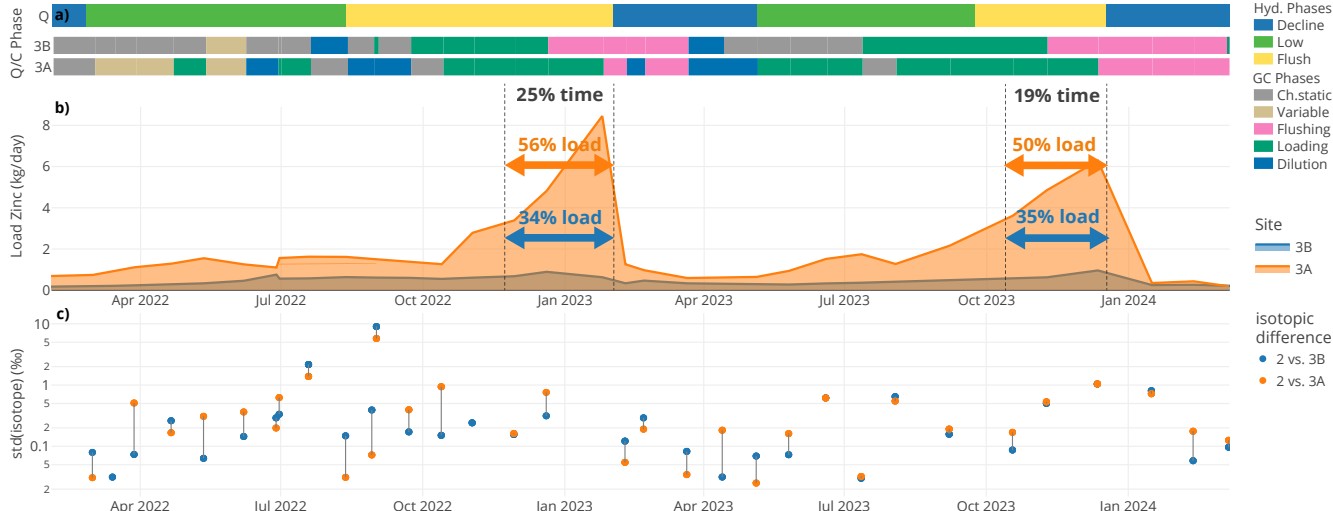

**Figure 4: Zn loads and isotopic similarity from February 2022 to March 2024 are shown for sites 3A (orange) and 3B (blue). a) hydrological phases (declining: green, low: yellow, flush: blue) and corresponding geochemical phases (e.g., loading, flushing, recession, dilution, and variable) for sites 3A and 3B (as in Fig. 3). b) Zn loads with peak load contributions labeled as percentages, indicating the proportion of total Zn export occurring in the 2-3 months preceding flushing. c) Standard deviation of isotopic concentrations (∂2H and ∂18O) to site 2. Difference between 3A and 3B marked as vertical lines. The relative ratio marks how much the water is similar (strong connection in flow field) or distinct (disperse paths) in the drainage system.**

The temporal synchronization of contaminant peaks across mine levels reveals a threshold-driven system governed by internal storage, episodic connectivity, and flow path structure. These processes collectively modulate metal(loid) transport and release, with site-specific patterns highlighting the role of delayed response and spatial heterogeneity in shaping contaminant export dynamics.

### 3.3 Dynamics in concentration-discharge relationships

Going beyond temporal and spatial trends, C-Q slopes (Table S3) provide insight into solute source proximity, mobilization timing, and hydrological connectivity (Knapp et al., 2020; Winter et al., 2021). Figure 5 presents diagnostic C-Q relationships for Cd, Zn, Fe and PLI at sites 2, 3A, and 3B, integrating hydrologically and geochemically defined phases. C-Q slopes (calculated for the entire sampling period) revealed very strong dilution patterns (b < 0) at sites 3A and 3B for Zn and Cd, which implies not only decreasing concentrations with increasing discharge but also a reduction in total solute loads compared

to the pre-flushing conditions. This suggests that at sites 3A and 3B incoming event water diluted the system more strongly
than metals were being mobilized, pointing to a depletion of readily exchangeable or previously accumulated solute pools.
One possible explanation is that during preceding low flow periods, metals accumulated locally but were not efficiently flushed
out during subsequent events, resulting in net declines in exported mass. In contrast, site 2 displayed a slope of -0.55, still
indicative of dilution but less pronounced, consistent with partial mobilization of stored solutes alongside dilution by incoming
recharge waters. These differences highlight that while all three sites show dilution-dominated behavior, the deeper sites (3A
and 3B) were characterized by stronger depletion, whereas site 2 retained evidence of ongoing solute mobilization events.
Within these general behaviors, we identified a "main pollution point", the moment of peak concentration coinciding with
minimal discharge typically seen during the hydrologically defined low flow periods, highlighting a critical window of
contaminant risk. Eventually these points are followed by a shift to high flow and low concentration marked by dilution.
Segment-scale C-Q analysis further clarified how internal storage and release differ among sites. Across all metals at these
main pollution points, $CV_c/CV_q$ ratios generally exceeded 0.5, and HI values consistently exhibited counterclockwise loops,
confirming that concentration variability is chemodynamic and governed by time-lagged storage-release dynamics, not
discharge magnitude alone. At site 3A, the combination of steep negative slopes and elevated $CV_c/CV_q$ values (up to ~6)
indicates deep, isolated pockets of solute-rich water undergoing intense build up during low flow and abrupt depletion upon
early stage flushing. Site 3B exhibited similar but less extreme patterns, suggesting pools that accumulate solutes during quiet
periods but are shallower or refreshed more frequently. In contrast, site 2 showed the weakest dilution yet remained
chemodynamic, supporting the interpretation that this nearer to surface channel receives a continuous supply of water, allowing
only partial flushing of stored solutes.
When evaluated alongside the timing of pollution potential points, the data reveal that each site expresses hot moments through
different mechanisms. At site 3A this is through rapid collapse of deep, enriched pools and at site 3B through more moderate,
short-lived spikes, and at site 2 through smaller concentration peaks. Collectively, these site-specific differences underscore
how the location, connectivity, and hydrodynamic activation of contaminant-rich zones govern release dynamics, highlighting
that standard outlet-based monitoring may overlook episodic contributions from deeper, disconnected compartments. This has
direct implications for the design of monitoring strategies and for timing interventions to capture or mitigate short-lived
contaminant pulses.

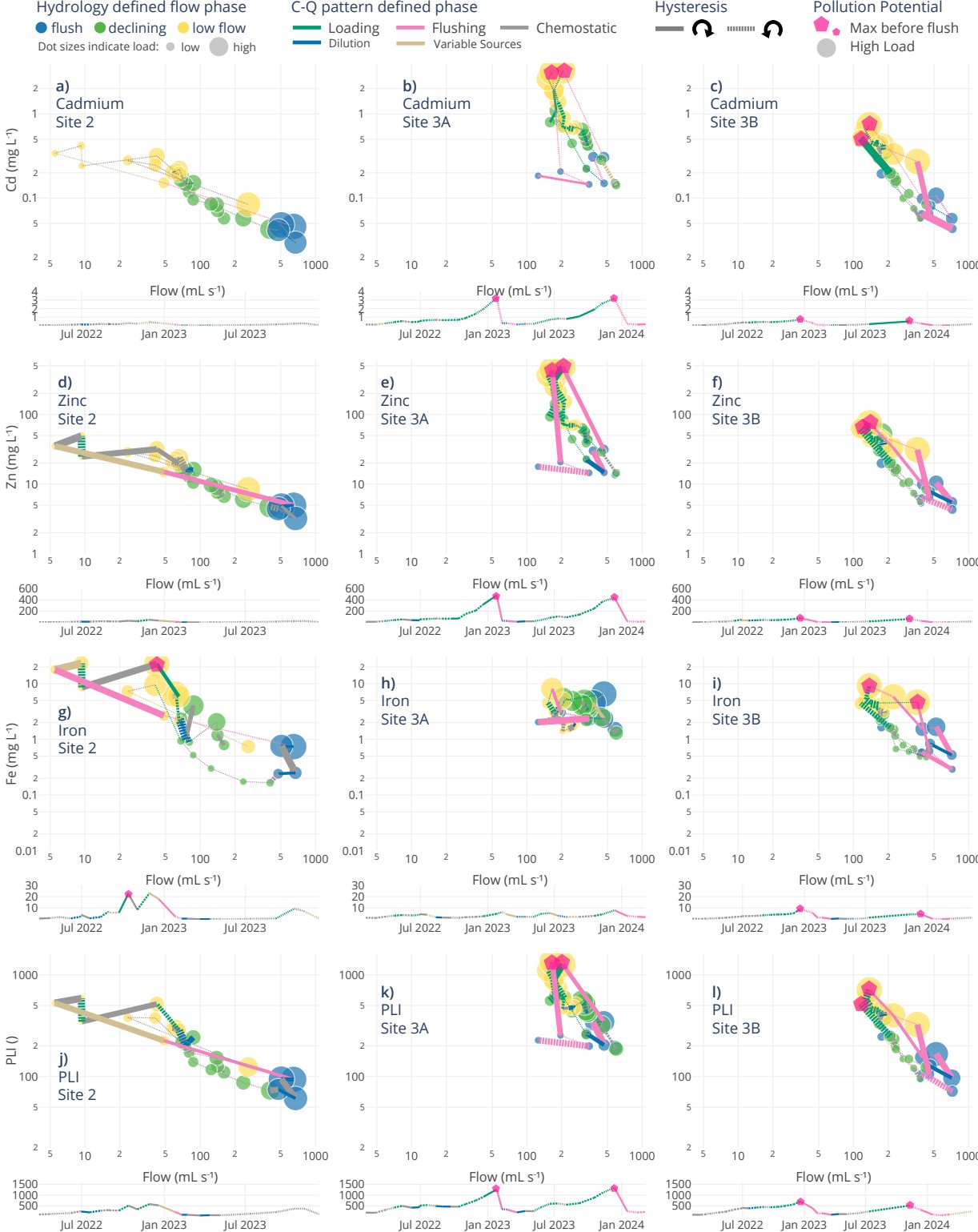


**Figure 5: Concentration–discharge (C–Q) relationships (log$_{10}$ scaled) for (a-c) dissolved Cd, (d-f) dissolved Zn, (g-i) dissolved Fe, and (j-l) Pollution Load Index at Site 2 (a, d, g, j), Site 3A (b, e, h, k) and Site 3B (c, f, i, l) during the three hydrologically defined phases (flush - blue, declining - green, and low - yellow). Bright pink pentagon-shaped points are the main pollution points (i.e., moments when high contaminant potential and high load is identified). Dot size represents the respective load. Under each C-Q plot, the dissolved metal concentration and PLI values are shown across a time scale. The geochemically defined phase from the C-Q patterns is represented through the color of the lines connecting each point for the C-Q plots with the line thickness corresponding to absolute hysteresis index (Zuecco et al., 2016) and is distinguished by its direction (negative or near-zero HI dashed, positive HI solid). This phase distinction is also shown for the time series with marked main pollution points.**

These short-term C-Q transitions observed across the sites map directly into the geochemically defined phases. The strongly negative C-Q slopes observed at sites 3A and 3B occurred during our defined loading phase before the onset of flushing. These segments reflect solute accumulation during low flow followed by a rapid decline in concentrations as connectivity begins to increase, consistent with a system that has accumulated solutes in lotic pathways but is not yet connected enough for flushing. As discharge begins to rise and thresholds are crossed, the system transitions into the flushing phase, marked by rapid release of solute-rich waters from lentic pools. These flushing periods were generally shorter-lived compared to loading periods. Following flushing, the system rapidly enters the dilution phase, where connectivity remains high, but solute stores become limited. Concentrations continue to decrease but with comparably less negative C-Q slopes than those seen during the loading phase, reflecting source-limited dilution in a partially lentic system. Notably, recent work has shown that C-Q slopes can diverge substantially between event-scale and long-term observations (Winter et al., 2024), underscoring the importance of analyzing C-Q relationships at short timescales using high-frequency data. During the later portions of events, we anticipated recession phases to emerge, but within our automated classification, chemostatic-like conditions were more prominent at sites 2 and 3A, pointing to similar characteristic behavior of these two phases in which the system transitions back toward reduced connectivity.

Importantly, while these classifications provide a coherent and mechanistic lens for interpreting short-term hydrogeochemical behavior, we view them as guiding tools rather than fixed or exhaustive categories. This aligns with recent discussions (e.g., Knapp and Musolff, 2024) emphasizing that C-Q based frameworks, even when combined with additional metrics, cannot capture all processes in complex subsurface environments. Thus, our framework serves as a structured interpretive aid highlighting dominant patterns and the threshold-driven transitions in this mine system, where lotic-lentic and fill-and-spill cycles jointly produce hotspots and hot moments of contaminant release.

### 3.4 Identification of high contaminant potential through spectrometric data

To complement this integrative view from the tri-weekly sampling and to zoom into the transition from loading to mobilization, we applied high-frequency monitoring with an in-situ UV-Vis spectrometer (collecting hourly measurements) and daily autosampling at site 2. At site 2, a drainage channel with rough bottom structure creates a specific dynamic flow environment where stratification, density contrasts, and throughflow coexist (Sanchez et al., 2025). This setting created a labile 2-phase

system: a low-density surface layer forming a hydraulically connected stream above a dense, metal-enriched bottom pool. Such dual compartments acted as both storage and release zones, with stratification intermittently buffering and then abruptly mobilizing contaminants. To capture these dynamics, we combined autosampler-based laboratory measurements with high-frequency spectrometric estimates of dissolved Cd (Fig. 6). This dual approach revealed that transitions from solute accumulation to flushing occurred within hours and much more rapid than what can be resolved by sampling alone, highlighting short-lived but significant windows of contaminant export.

All monitoring methods consistently showed Cd buildup during low and declining flow (July – December), followed by sharp concentration drops at the onset of flushing conditions. These shifts reinforce the role of threshold-based activation and transient hydrological connectivity in controlling solute mobilization. In addition to these processes, the dynamics are also consistent with gradual solute accumulation under density-stratified conditions followed by abrupt dilution conditions once connectivity is established. Notably, stratification observed visually in mid-October 2022 to late January revealed Cd concentrations in bottom grab samples (positioned near the channel base) more than two orders of magnitude higher than top grab samples (collected at the surface). The grey polygon (i.e., grab sample corridor) in Fig. 6a delineates the concentration range captured by these paired depth-integrated manual grab samples, as well as the spectrolyzer results over the low flow period, providing a window into the presence of density-stabilized, solute-rich bottom waters. These findings suggest temporary solute traps forming during quiescent conditions, a phenomenon also reported in mine systems with intermediate density layering that can be rapidly flushed upon reactivation (Mugova and Wolkersdorfer, 2022; Mugova and Wolkersdorfer, 2024).

Figure 6b illustrates how phase-specific C-Q relationships for dissolved Cd evolved during a six-week period leading up to the major flushing event in February 2023. Hourly data revealed shifting mobilization regimes, with clear transitions between chemostatic and chemodynamic behavior, just before hydrological shifts. Each C-Q panel corresponds to a high-resolution sub window from the flow time series, capturing short-lived events with discrete geochemical responses. This highlights how although during small or intermediate increases in flow, characteristics of chemostatic conditions may prevail, in a short time span these conditions can change and be initiated from accumulated solute pools, especially under stratified or semi-stratified conditions where density-driven segregation creates temporary storage zones. During low to moderate flow conditions, even minor discharge increases can lead to the buildup prior to contaminant mobilization when residual sources remain available, eroding micro-stratification and reconnecting isolated pools with high pollution potential. In contrast, during high flow conditions, when temporary storages are already exhausted, these pools may be heavily mobilized, such that a variability of dilution, loading, and recession typically dominate the C-Q relationship (Fig. 6b, 'Mobilization and eventual beginning of dilution' panel). Solute transport regimes transition within just a few days to weeks as discharge fluctuates, underscoring the dynamic connectivity of source zones and complements the broader patterns seen across sites and metals in Fig. 5. These findings reinforce that episodic hydrological forcing can lead to metal-specific, rapidly evolving export regimes that cannot be captured by temporal sampling alone. Our results further illustrate how event-scale monitoring captures transient transport processes that could be obscured in analyses based solely on long-term, low-frequency data.

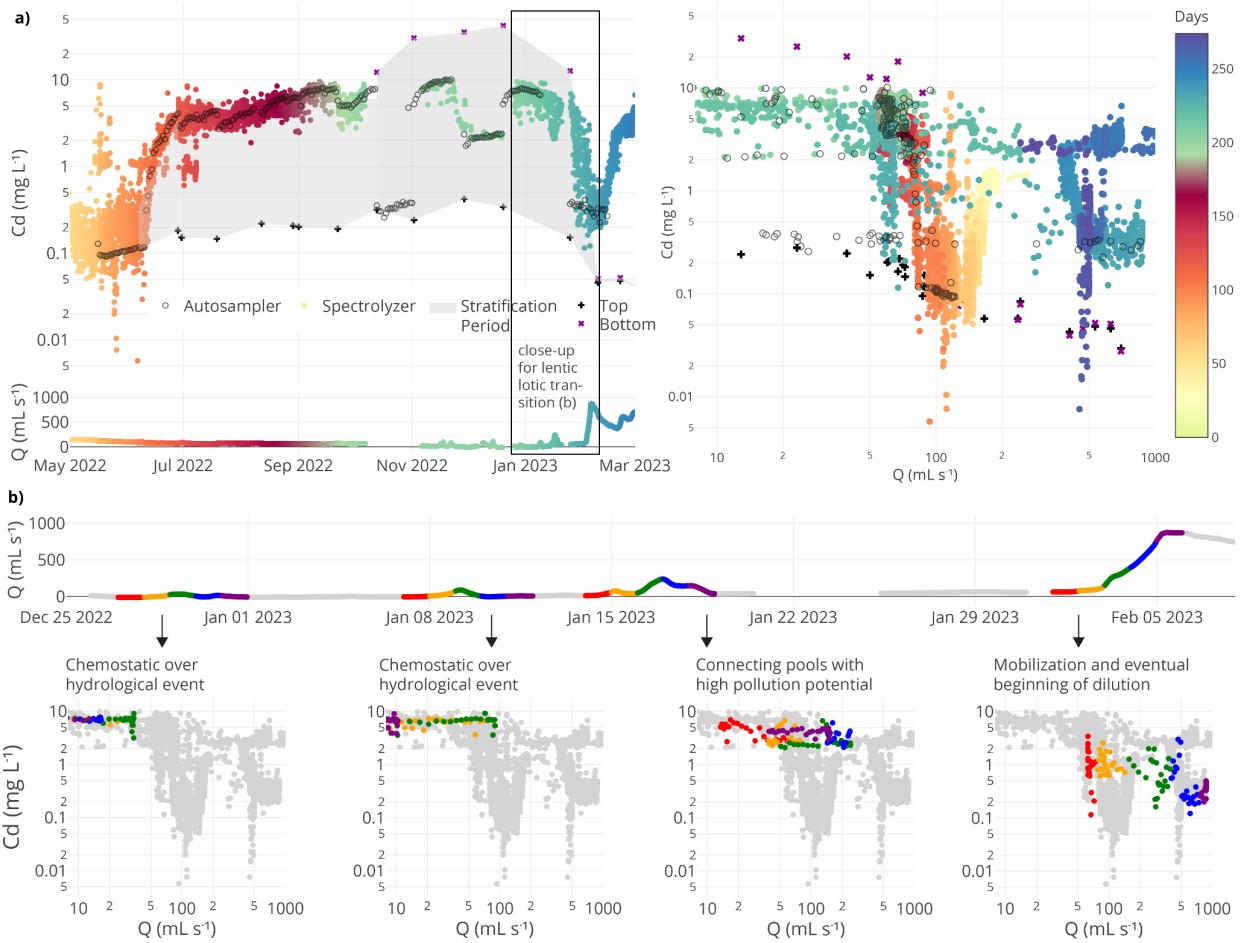


**Figure 6: (a) Top right panel: Dissolved Cd concentrations (log$_{10}$ scaled) at site 2 measured using an autosampler (daily values in**
**open circles), the spectrolyzer instrument (hourly values in colored circles), and manual sampling (bottom - purple crosses; top -**
**black crosses). The grab sample corridor (grey polygon) represents the concentration range in which stratification was evident based**
**on all sampling methods (until stratification collapse). The spectrolyzer measurements are colored by sampling day to indicate**
**temporal regression. The black box with the annotation of close-up for lentic lotic transition highlights the six-week window analyzed**
**in detail in panel b. Bottom panel: The time series for the water discharge matching with the spectrolyzer color scheme. Top left**
**panel: C-Q relationship (log$_{10}$ scaled) for dissolved Cd following the same spectrolyzer color scheme as the right panel. (b) Top panel:**
**The time series of the six-week window for the discharge matching with the spectrolyzer color scheme. Bottom panels: Phase-**
**resolved C-Q patterns (log$_{10}$ scaled) for dissolved Cd over the stratification-collapse and lentic-lotic transition. Each small C-Q panel**
**shows hourly Cd-Q data for a 5-day segment, with points colored by each day (day 1 - red, day 2 - orange, day 3 - green, day 4 -**
**blue, day 5 - purple). C-Q plots highlight progressive phases in the transition between lentic to lotic or fill to spill conditions. After**
**the first two hydrological events showing chemostatic characteristics (2022-12-27 to 2022-12-31 and 2023-01-07 to 2023-01-11), the**
**third event hints to connecting pools with high pollution potential (2023-01-14 to 2023-01-18), which are eventually mobilized in the**

**fourth and strongest breakthrough event mobilization (2023-02-01 to 2023-02-05). These panels are linked to their respective periods in the hourly discharge time series.**

The combined use of autosampler and high-frequency spectrolyzer data offer synergistic insights, such that the former anchors the dataset in analytical accuracy, while the latter captures transient solute behavior and enables time-resolved analysis of flow-phase transitions. This integrated perspective, alongside zooming into small time windows of C-Q responses, is critical for detecting and characterizing hot moments, in which brief but disproportionate pulses of metal export can dominate annual metal loads. Our findings emphasize that stratified pools stabilized during low flow can be rapidly activated within a short window during fill-and-spill (McDonnell et al., 2021) or lentic-lotic cycle transitions (Shaw et al., 2020), reinforcing the need for depth-aware, event-sensitive monitoring to anticipate episodic contaminant risks in complex mining systems.

### 3.5 Implications for contaminant pollution remediation

Our findings highlight that contaminant mobilization in abandoned mine systems is not primarily driven by storm intensity or seasonal high flows, but by internal hydrological thresholds and episodic connectivity between stored contaminant pools and the active drainage network. The observed lag between peak accumulation and flushing, followed by rapid load collapse, suggests that predictive assessments must incorporate not just hydrometeorological variables but also the internal memory of the system.

These dynamics expose critical blind spots in current water quality monitoring and regulatory frameworks. Existing benchmarks, such as German sediment quality guidelines (e.g., 800 mg/kg Zn in suspended materials) (Bundesamt für Justiz, 2016) and EU background dissolved Zn concentrations (1–35 µg/L) (Munn et al., 2010; Comber et al., 2008) overlook the timing and intensity of short-lived, high-risk release events from underground contaminant reservoirs. Figures 5 and 6 illustrate this clearly. Figure 5 provides a system-scale diagnostic, showing how metals and PLI evolve across sites and flow phases, with highlighted hot moments pinpointing pulses that carry disproportionate contaminant loads. Figure 6 then zooms in at high temporal resolution, capturing the collapse of stratification and lentic-lotic transition that triggered a breakthrough event. Therefore, the hot moments of release, which arise during hydrologically quiet intervals rather than extreme events, may represent important windows of strategic intervention.

While this study focuses on the Reiche Zeche mine, the fill-and-spill dynamics we observe are likely widespread across hydrologically complex, mining-impacted systems, especially in porous systems with variable subsurface connectivity, stratified drainage zones, or episodic flow regimes. Similar mechanisms may be active in karst aquifers, tunnel-fed drainages, or engineered infrastructure where discrete contaminant pools are intermittently connected to surface outflows. This shift from peak flow emphasis in such systems toward detecting internal system thresholds supports a more proactive, precise, and strategic path for better timed remediation. Effective mitigation depends on anticipating these moments before widespread flushing, when contaminant concentrations are high but spatially contained.

At site 3A, located near the central drainage adit and consistently exhibiting the highest PLI values, we observed Zn loads up to 8.4 kg/day prior to a major flush event (mean 1.9 kg/day). This site contributes only 0.06% of the overall water to the outlet of the overall adit Rothschönberger Stolln, but 1.3% of the Zn load in average (LfULG, 2014) with the few days before flushing accounting for about 50% of the annual load from flux-based metrics. While a more detailed monitoring using a spectrolyzer at site 3A would have been advantageous, this comparison underscores that substantial contaminant fluxes can accumulate and be released from within the mine system itself, often remaining undetected by conventional downstream monitoring. By combining C-Q relationships and hysteresis analysis, our approach pinpoints internal hotspots and identifies hot moments of high mobilization risk, advancing a framework to guide targeted monitoring and early-warning systems. Beyond this study, these findings highlight the broader relevance of upstream diagnostics for understanding contaminant behavior in legacy mine settings and support the need for spatially resolved, phase-sensitive strategies for remediation planning. The latter could include specific small-scale treatment systems near to the actual mobilization hotspots in the legacy mines.

While our results emphasize the value of identifying internal hotspots and hot moments, remediation strategies involve clear trade-offs. End-of-pipe treatments (e.g., treatment at mine outlets) offer practical advantages because they can operate as a single, accessible location and do not require detailed knowledge of internal connectivity, but they may miss short-lived contaminant pulses generated upstream. Source-proximal or hotspot-focused interventions can intercept highly concentrated releases earlier, yet risk overlooking additional, undetected hotspots in hydrologically complex systems. Given that contaminant mobilization is highly dynamic and rarely captured in current monitoring frameworks, an effective remediation strategy likely requires combining system-scale and end-of-pipe safeguards with targeted upstream diagnostics to balance feasibility with responsiveness to episodic release events.

## 4    Conclusions

This study demonstrates that contaminant mobilization in abandoned mine systems is controlled not by steady seepage but by episodic shifts in internal hydrological connectivity. Across the Reiche Zeche mine, low flow and pre-flush phases were shown to concentrate dissolved metal(loid)s in poorly connected storage zones, with subsequent reconnection triggering sharp but short-lived contaminant releases. Event-scale C-Q relationships and indices reveal that such hot moments of export account for a disproportionate share of annual metal loads, emphasizing the need to move beyond traditional outlet-based monitoring. Our findings highlight three key insights: First, low flow periods represent high risk intervals of solute accumulation, challenging assumptions that contaminant risk is greatest only during floods or peak flows. Second, site-specific C-Q dynamics demonstrate that contaminant export is shaped by rapid transitions between hydrogeochemical phases, capturing how internal hotspots formed during low flow evolve into hot moments of connectivity-driven release. Third, targeted monitoring of connectivity threshold provides a basis for early warning and site-specific and near-source interventions. By identifying internal hotspots and the timing of mobilization events, this work establishes a transferable framework for diagnosing contaminant risks in legacy mine settings. These insights support a shift toward event-sensitive, near-source remediation

strategies that prioritize internal system dynamics, offering more efficient and scalable alternatives to conventional end-of-pipe treatment.

## Author contributions

**A.A.S.**: Conceptualization, Methodology, Formal analysis, Investigation, Data curation, Writing-original draft, Writing-review and editing. **M.P.L.**: Conceptualization, Supervision, Validation, Writing: review and editing. **S.A.**: Spectral data analysis, online UV-Vis Spectrometer data curation. **S.H.**: Validation, Writing: review and editing. **C.J.**: Conceptualization, Supervision, Hydrological and hysteresis analysis, Data visualization and curation, Original Writing-original draft, Writing: review and editing.

## Data Availability Statement

The dataset supporting this study is openly available via the B2SHARE data repository under the title LegacyMine_HydroGeo: Dataset on the geochemical and hydrological dynamics in a historic mine system (Sanchez et al., 2026). It includes high-resolution geochemical, isotopic, hydrological, and spectrometric data collected from the Reiche Zeche mine over a two-year monitoring period. The dataset can be accessed at https://doi.org/10.23728/b2share.566z6-gtm46 and the geochemical phase classification suite is available at https://doi.org/10.5281/zenodo.18462921.

## Competing interests

The authors declare that they have no conflict of interest.

## Acknowledgements

This research was conducted as part of the project "Source Related Control and Treatment of Saxon Mining Water" and was funded by the Dr. Erich-Krüger Foundation. We extend our gratitude to Dr. Alexander Pleßow for his support with laboratory equipment. Special thanks to Prof. Helmut Mischo and Stephan Leibelt for their training and access to the Reiche Zeche mine, as well as to Dr. Andreas Kluge and Dr. Nils Hoth for their invaluable assistance in initiating work on the mine system. We also wish to acknowledge the laboratory team—Thurit Tschöpe, Marius Stoll, Claudia Malz, Eva Fischer, and Lena Grundmann—for their help with sample measurements, as well as Karl Haas and Lena Herzig for her assistance in collecting mine water samples. We also thank Prof. Erwin Zehe and his team at KIT Karlsruhe for lending the ISCO autosampler.

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
