# Peer review of "Hotspots and Hot Moments of Metal Mobilization: Dynamic Connectivity in Legacy Mine Waters"

_EGUsphere, 2025_

## Author Response (AR1)

**Institut für Bohrtechnik und Fluidbergbau**
Professur für Strömungs- und Transportmodellierung in der Geosphäre

[Figure]

To the Editor and Referees of
Hydrological and Earth System Sciences

| | |
|---|---|
| Bearbeiter: | **Jun. Prof. Dr. Conrad Jackisch** |
| Anschrift: | Werner-Arnold-Bau, Agricolastraße 22, |
| Telefon: | 09599 Freiberg |
| E-Mail: | 03731 39-2681 |
| Homepage: | Conrad.Jackisch@tbt.tu-freiberg.de |
| | tu-freiberg.de/fakult3/ibf |
| Datum: | 06.12.2025 |

**Response to editor and referees**

We thank the editor for the opportunity to submit a revised manuscript and the referees for their thoughtful evaluation on our manuscript "Hotspots and hot moments of metal mobilization: dynamic connectivity in legacy mine waters". We are very grateful for the important clarifications the referees requested for, which we addressed when revising the manuscript. Briefly, the major changes that we made were:

1. Revised analysis hysteresis and interpretation – We considerably revised our treatment of hysteresis. Following the referee's guidance, we recognized that hysteresis direction cannot be interpreted independently of the underlying C-Q behavior and that for an enhanced process-based interpretation of our C-Q patterns, hydrological and geochemical context should be included. We therefore removed detailed interpretation based on hysteresis direction loop type. Instead, we combine multiple indices related to C-Q relationships (C-Q slopes, CVc/CVq ratios, hysteresis analysis), as well as wetness conditions, and PLI and discharge trends to differentiate between chemostatic and chemodynamic conditions and to support the development of our geochemical phase framework. Section 2.6.4 has been simplified accordingly, and interpretations in the results and discussion section have been updated. Despite the revisions of the hysteresis analysis, the main findings remained.

2. Clarified terminology ("load" and "loading") – To avoid confusion between concentration increases and flux terminology, we now explicitly distinguish loading (temporal increases in solute concentration or accumulation) from "load" (flux, C*Q). This was stated early on in the abstract of the manuscript when these terms were first used.

3. Improved geochemical phase definitions and visualization – We revised Section 2.6.5 to clearly describe how geochemical phases were derived from point-to-point C-Q slopes, CVc/CVq ratios, hysteresis indices using methods from Lloyd et al., (2016), Zuecco et al. (2016) and Roberts et al. (2023), trends in PLI and discharge, and hydrological flow context, rather than solely from hysteresis. To address the referee's concern that some phases share similar slope directions, we now state that loading and dilution both may have negative relationships, but that they are distinguished by differences in flow state, and solute availability, as shown in our revised conceptual Figure 2 and phase timeline in Figures 3, 4, and 5. The amendments helped to clarify how the phases relate to fill-and-spill dynamics, lotic-lentic cycling, and hotspot/hot moment behavior.

4. Streamlined and refocused the introduction – To improve readability and focus, we condensed the introduction to emphasize the key challenges, knowledge gaps, and motivation for the study. Conceptual ideas and former Figure 1 (now Figure 2) were relocated to the Methods section. This helped to reduce complexity and improve the story flow.

5. Condensed and clarified aims, methods, and study rationale – We simplified the aims and objectives into one overarching research question with three concise objectives. In parallel, we removed excessive justification in the methods section, ensuring the focus remains on this study's design and findings.

Beyond this general plan, here is the detailed response to the referees comments:

**Referee #1**: In their manuscript entitled „Hotspots and Hot Moments of Metal Mobilization: Dynamic Connectivity in Legacy Mine Waters", Sanchez et al., (2025) analyzed metal(loid) pollution dynamics in a mining-affected area. Based on an extensive monitoring program and using C-Q relationships, the authors reveal the high spatial and temporal variability of hydrological metal(loid) transport. The results of this study are quite interesting and therefore likely to be a highly valuable contribution for the readers of HESS. Furthermore, the study is well-written and comprises clear and appealing figures. I therefore recommend considering publishing this study in HESS. However, I do not entirely agree with some of the interpretations of the results, especially regarding the hysteresis analysis, but also with the "loading" term and the definition of phases. Therefore, additional consistency checks and potential revisions to certain parts of the discussion may be necessary, as detailed below.

[Figure]

[Figure]

[Figure]

Thank you very much indeed for your thoughtful evaluation of our manuscript. We see that we have caused some haze with insufficiently crisp wordings piling up to slight inconsistencies. This has now been addressed along with the lines presented below.

*Hysteresis analysis and interpretation*

A clockwise hysteresis coupled to an enrichment pattern can imply something very different than a clockwise hysteresis coupled to a dilution pattern, and vice versa. Together with an enrichment pattern, clockwise hysteresis indicates a faster reaction of concentrations (C) as compared to discharge (Q), in the form of a faster increase of C, a faster recovery after the peak, or both. Consequently, counterclockwise and enrichment indicate a slower reaction of C as compared to Q. Instead, dilution and clockwise hysteresis imply a slower reaction of C as compared to Q, in the form of a delayed decrease in C, a slower recovery/increase after the peak, or both. Counterclockwise hysteresis and dilution imply a faster response of C compared to Q. As this may not be entirely intuitive, I have added a hand-drawn sketch to illustrate this for dilution patterns (as these were the primary patterns reported). I hope it is readable.

Given these different C responses for the same hysteresis pattern, one must be very careful and avoid interpretations based on enrichment patterns from other studies. So one cannot generally say that "Clockwise loops suggest rapid mobilization from proximal sources", as stated in the manuscript and later on built upon that statement for further interpretation. This needs to be carefully checked throughout the manuscript, and interpretations based on this need to be re-evaluated.

We thank the reviewer for the careful explanation of how hysteresis direction must be interpreted together with the underlying C-Q behavior. We fully agree that loop direction alone cannot be used mechanistically and that the meaning of clockwise or counterclockwise depends fundamentally on whether concentrations increase or decrease relative to discharge. We appreciate the sketch provided, which helped clarify this distinction.

In revising the manuscript, we carefully re-evaluated our own data in light of this comment. We found that while hysteresis indices (Lloyd et al., 2016; Zuecco et al., 2016; and now included Roberts et al., 2023) were useful to compute, they did not provide the main mechanistic insight into solute sources or timing. This is largely because our system is dominated by dilution-type C-Q behavior, in which loop direction varies but does not meaningfully differentiate processes or source distances. As such, we agree that overstating the interpretive power of hysteresis would be misleading and as Knapp and Musolff (2024; https://doi.org/10.1002/hyp.15328) suggested, C-Q tools should be used with caution and not relied on as the sole method for analyzing concentration and discharge time series but rather in conjunction with other information, such as hydrological and biogeochemical conditions.

For this reason, we have adjusted our manuscript so that hysteresis analysis is included only as one component of a broader multi-metric framework rather than a standalone interpretive tool. Specifically, we calculated C-Q slopes (b), CVc/CVq ratios, hysteresis indices from Lloyd et al. (2016) and from Zuecco et al. (2016), and HARP (Hysteresis Area, Residual, and Peaks) from Roberts et al. (2023). These metrics were integrated into an algorithm, point-to-point classification scheme that assigned each time segment to one of our geochemical phases. Hysteresis indices contributed to this classification, but they did not drive our process interpretation. Instead, the dominant interpretive weight came from the combination of C-Q slope direction, CVc/CVq variability, PLI and discharge dynamics, water availability index values, and lentic-lotic and fill-and-spill connectivity dynamics.

Accordingly, in the revised manuscript: (1) All mechanistic interpretations solely based on hysteresis loops have been removed. (2) Detailed methodology on hysteresis calculations are moved to the SI. (3) The methods section (2.6.4 and 2.6.5) has been updated to clarify why hysteresis indices were calculated to support classification but were not used as a primary interpretive tool. (4) The combination of C-Q slopes, CVc/Cvq ratios, and hydrological context proved more informative than hysteresis alone for defining our geochemical phases and for resolving short-term transitions in our mine system and we clarified this. (5) Figure 2 has been updated to show how the C-Q patterns observed in our system align with our fill-and-spill and lentic-lotic cycling concepts.

Additionally, while I agree that hot moments observed for 3A are striking, the dynamics for 2 and 3B are less pronounced. They appear to show slow and steady buildup of concentrations, for example, due to an increasing water age during low flow, and a rapid dilution with incoming (younger and less exposed) event water (as also described in lines 465-466). I do not see much of the mobilization pattern or threshold behavior here, which is constantly referred to in the manuscript. Overall, I recommend a clearer differentiation between the potentially driving processes at different sites and a reevaluation of the potential underlying processes.

We appreciate your observation. We agree that the strength of threshold-driven mobilization differs across sites, with site 3A showing the clearest evidence. However, we emphasize that site 2, exhibiting gradual solute buildup during low flow, still revealed threshold-like dynamics when viewed in detail. These features include stratification and density-driven

Institut für Mineralogie,· Brennhausgasse 14,· 09599 Freiberg · www.tu-freiberg.de/fakult3/

storage, abrupt concentration drops at the onset of flushing, and phase-specific hysteresis behavior. These features indicate that threshold exceedance processes were also operating at site 2, though modulated by site-specific hydrodynamics. In the revised manuscript (lines 592-597), we clarified that threshold-driven behavior was most pronounced at site 3A, while at site 2 and 3B additional processes (e.g., dilution by younger recharge waters, stratified storage) also played a role as well. Since obviously, a more detailed monitoring at site 3A (using a spectrolyzer there too) would have been advantageous but simply has not been done. We added this aspect in the discussion too.

*Loading*

Finally, I got a little confused by the terms 'load' or 'loading'. My initial interpretation was 'load' in terms of C * Q. Only when it came to explaining the PLI did I understand that it is about concentrations, which makes a difference in how to interpret the results. To avoid this confusion, I suggest choosing a different term or explaining this difference in terminology early on in the manuscript.

Thank you for drawing our attention to this ambiguity. In the revised manuscript, we explicitly defined "load" as flux (C*Q) and state that the term "loading" is used to describe temporal concentration accumulation. This distinction is now introduced early on in the manuscript when these terms are first presented and is applied consistently throughout the manuscript.

*Hydrogeochemical phases*

I find the naming and visualization of 'phases' somewhat confusing. In the sense of the directional C-Q relationship, what is here defined as 'Loading' and 'Dilution' describe the same negative relationship between C and Q. The same applies to what is called 'flushing' and 'recession', which both describe a positive relationship between C and Q. While it might make sense to divide these phases also into those of rising and falling discharge, their common C-Q relationship should be made clearer in the text and in the figures, for example by coloring the same C-Q relationships in a more similar way. Also, I did not entirely understand how that was calculated. If it is based on the C-Q relationships, was it calculated using a moving average, or how were the phases separated? I might have missed it, but the manuscript would benefit from stating that clearer in the methods.

We thank the reviewer for this highly constructive comment. We now see more clearly why the earlier naming of "loading/dilution" and "flushing/recession" caused confusion, because in a classical C-Q interpretation these pairs indeed share the same slope direction (negative or positive). We appreciate the opportunity to clarify this, as our revised conceptual framework is now better aligned with the actual hydrogeochemical functioning of the mine system.

In our system, C-Q slope alone cannot distinguish the dominant process. We therefore do not classify phases based solely on the sign of the slope, but rather we use a combination of C-Q slope, hysteresis indices and metrics (methods from Lloyd et al., 2016; Zuecco et al., 2016 and Roberts et al., 2023), $CV_c/CV_q$ ratio, PLI and discharge trends, and water availability index values. These parameters on a temporal scale for each site are now shown in the SI (Fig. S3) since this aided in our automated classification. We tie these indices with our hydrological concepts (fill-and-spill activation and lentic-lotic state) to differentiate between distinct mechanistic states.

While slope sign alone would collapse the phases into two categories, our system exhibits temporally distinct process regimes that may share slope direction but differ mechanistically:
1. Loading: Segments with increasing PLI values, negative C-Q slopes, and negative hysteresis during which flow is at its low or increasing were classified as loading. These conditions reflect moments where water resides long enough in isolated pools or channelized pathways for solute stores to accumulate. These segments correspond to the filling state before threshold activation.
2. Flushing: Segments with initially high PLI values which lower as discharge increases, positive C-Q slopes, and positive hysteresis were classified as flushing. These are short time windows where solute-rich lentic layers spill and mobilize accumulated solutes as connectivity rapidly expands. This aligns with threshold exceedance and the activation of previously disconnected domains.
3. Dilution: Segments with variably high flows and declining PLI values, and relatively high $CV_c/CV_q$ ratios with positive hysteresis were classified as dilution. Here, solute concentrations decrease due to mixing depleted lentic waters and less solute-rich flow. Connectivity persists but source reservoirs become progressively exhausted.
4. Recession: Segments with lowering flow and stable or slightly declining PLI trends, very low $CV_c/CV_q$ ratios, and low water availability index values were classified as recession, These segments typically occurred during periods of declining flow when connectivity contracts and solute exchange with source zones is limited. While manual inspection would qualify some of the phases as recession, the automated classifier did not classify any segment as such. We however left the class definition, because we attribute this non-detection with the coarse temporal resolution of our data.
5. Chemostatic: Periods where flow slightly varied but PLI, C-Q slope, and $CV_c/CV_q$ ratios remained relatively stable with low $CV_c/CV_q$ (< 1) and flat C-Q slopes, and low hysteresis indices were identified as chemostatic. These

Institut für Mineralogie,· Brennhausgasse 14,· 09599 Freiberg · www.tu-freiberg.de/fakult3/

episodes occurred during sustained connectivity when reactive surfaces remain buffered and concentrations change minimally.

6.  Variable sources: Segments that did not match the characteristic patterns of other phases, typically showing relatively stable flow and PLI trends with mixed or ambiguous changes in C-Q slope, $CV_c/CV_q$ ratios, and hysteresis were classified as variable sources. These segments indicated solute dynamics driven by processes other than flow magnitude alone.

In the revised Methods (Sections 2.6.4-2.6.5), we now explicitly describe the workflow:

-   Phases were first determined by manually observing the point-to-point changes over consecutive sampling intervals for each site
-   For each segment, we calculated the C-Q slope and $CV_c/CV_q$ ratio using a five-point rolling window, as well as the hysteresis index values calculated on the time window surrounding each segment. These metrics were integrated into a hierarchical rule-based classification algorithm in which each segment was assigned a confidence score (0-1) based on how strongly its C-Q slope, $CV_c/CV_q$ ratio, and hysteresis behavior matched characteristic patterns for each phase. Phases were evaluated in priority order (flushing, loading, chemostatic, dilution, recession, and variable), with the first phase whose rules triggered being selected as the dominant phase of that segment.

Thus, loading and dilution may share similar slope behavior but differ in hydrological context and C-Q arrow orientation, which is now clearly shown in the phase timeline provided in Figures 3, 4, and 5.

**Line-to_line comments**

1. L21-24: This sentence is hard to understand. I suggest splitting and rewriting it.

Thank you for highlighting this. We see how the phrasing could be confusing. We split and rewrote the sentence in a more straightforward and easy to understand way.

2. L39: I am not sure 'discrete' is the right term here. As the measurement points in the study are discrete as well.

Thanks for noting this. We agree that discrete could be misleading in this context. We replaced this term with "low-resolution" to better convey that standard monitoring provides infrequent measurements and limited temporal coverage.

3. L61: Readers would benefit if this was explained a little more in detail.

We're grateful for this observation. We expanded the sentence to clarify what is meant by "lagged", "low-pass filtered", and "threshold-dependent" in lines 69-72. The revised text explains that near surface signals are delayed by slow percolation, short-term variations are dampened by storage effects, and responses often occur only after thresholds of connectivity or storage are exceeded.

4. L65: adding 'e.g.'? Here and in other exemplary citations as well?

Thanks for this suggestion. We agree that the citations presented serve as examples rather than an exhaustive list. Accordingly, we revised the sentence to include "e.g." before the citations to clarify this point.

5. L71: I find it a little hard to imagine what exactly the measurement points look like, if it is not a surface catchment. Maybe some photos, even if only in the SI, would help the readers to imagine the right thing here.

Thanks for raising this point. We understand that it may be difficult to imagine how the underground sampling points look like. We added photos of sites 1, 2, 3A, and 3B with the dates noted in which they were taken in the SI and mentioned this in the manuscript in lines 142-143.

6. L87-90: I suggest adding Musolff et al. (2015).

Thanks for this suggestion. Given that C-Q relationships have been applied in a wide range of studies, we have missed to add this important and clarifying reference. Since some of your conceptual suggestions appear to be well-aligned with this study, we revised the hysteresis analysis accordingly and added the citation.

7. L91-92: Only for enrichment patterns. See my comment above.

Institut für Mineralogie,· Brennhausgasse 14,· 09599 Freiberg · www.tu-freiberg.de/fakult3/

Thank you for highlighting this important nuance. We agree that our original statement ("clockwise loops suggest rapid mobilization from proximal sources…") was an oversimplification. We revised the manuscript in section 2.6.3 to clarify that hysteresis direction must be interpreted in the context of the underlying concentration-discharge regime. We also revised the conceptual representation of C-Q relationships in the context of our geochemical phases in our new Fig. 2 to avoid any misinterpretations.

8. Figure 1) Overall, the figures in the manuscript have a very high quality. Still, I have some suggestions for improvement:

c) Sorry to be a little peaky, but conceptually this is not entirely correct. For example, the lower dark blue dot ('flush') describes the same Q level as the upper green dot that describes 'declining flow'. The same applies to the upper yellow and lower green dots.

We appreciate this being highlighted. We decided to remove this subfigure to avoid overlapping discharge values across phases and replaced it with a revised version of Fig. 2. This now shows the combination of how fill-and-spill dynamics may present C-Q behaviors (Fig. 2a) which then help to develop our geochemical phases that incorporate this, and also hysteresis loops and hotspot and hot moment dynamics. We hope this provides more conceptual clarity.

d) I suggest not using numbers a-d twice in the figure, but instead using other numbering for the subfigures within 1d).

Yes. We decided to simply remove the numbering for the subfigures to avoid confusion.

9. L121: Does 'high-resolution' refer to spatial, temporal, or both?

Thanks for pointing this out. "High-resolution" refers to both spatial and temporal. We decided to remove the placement of this term here to make this section more concise.

10. L270-276: I suggest adding the equation here. I assume it is Log10(C) = a + log10(Q) ^b ?

We agree. The equation is Log10(C) = Log10(a) + b*Log10(Q) and we added it to this section in 291, which makes sense since we include the equation for PLI.

11. L273-274: b=0 does not necessarily imply that concentrations are stable. While this is often the case, it could also be that concentrations vary independently of Q. For this reason, Musolff et al. (2015) combined slope b with the $CV_C/CV_Q$ to test whether concentrations are stable. Hence chemostatic means b ~ 0 and $CV_C/CV_Q$ <<0.5. This is not only true for that figure, but for the general interpretation of results. I recommend applying the $CV_C/CV_Q$ to all metal(loids) to assure that b=0 can actually be described as chemostatic behavior.

Thank you for highlighting this important nuance. We fully agree that b ~ 0 alone does not imply chemostatic behavior, as concentrations may still vary independently of discharge. Following this suggestion, we calculated CVc/CVq for all metal(loid)s at the four sites. Because our study uses C-Q tools not to label segments purely "chemostatic" or "chemodynamic" but to inform geochemical phase classification, we clarify that the interpretation of b and CVc/CVq is adapted to the specific flow-path structure of the mine system. The Musolff et al. (2015) framework provides the quantitative foundation for assessing variability, but the resulting values are interpreted within our geochemical phase-based scheme, which incorporates additional hydrological context (i.e., water availability index, lentic-lotic transitions). We added this clarification to the methods in Section 2.6.3.

12. L276: I would argue that a positive slope does not indicate 'increased hydrological connectivity', but an increased mobilization of solutes during increased hydrological connectivity.

We agree with this statement and amended the text accordingly.

13. L278-290: I suggest adding the equations here as well, so that readers can understand how this index was calculated.

We agree that the equations used to calculate the hysteresis index values from the methods of Lloyd et al. (2015) and Zuecco et al. (2015) (and also newly added from Roberts et al., 2023) should be included. Since the hysteresis analysis was not our main focus in the manuscript we moved the detailed methodology to the SI with these equations added.

14. L306: I suggest citing the python packages used.

Institut für Mineralogie,· Brennhausgasse 14,· 09599 Freiberg · www.tu-freiberg.de/fakult3/

Yes, this is fair. We revised the methods section to include citations for the Python packages (pandas and plotly libraries) used.

15. L323-325: Couldn't it also be that the water just needs a little longer to percolate down to these deeper layers?

Thanks for this good perspective. While delayed percolation through the subsurface could in principle contribute to the lag between surface conditions and increased discharge, our observations suggest that deeper layers in this system generally respond rapidly, with streamflow dynamics at depth aligning closely with overall discharge. We therefore interpret the observed lag not as a simple percolation delay, but as evidence of threshold-based fill-and-spill dynamics within vertically structured storage zones. We revised the discussion to acknowledge delayed percolation as a possible mechanism, but emphasized that our data support threshold-driven connectivity as the dominant explanation in this setting in Section 3.1.

16. Figure 3: Again, very nice figure. For the coloring of phases, see my comment above. Also, the light and dark blue colors are hard to distinguish, especially if the lines are dashed.

Thank you for the compliment and for the suggestion. We addressed this by making loading and dilution similar hues and flushing and recession similar hues. We also made sure that the color of the dashed lines are more clearly visible and all colors presented in the figure are clearly distinguishable.

17. L361: I would be interested to see the dynamics of the other metal(loids) measured as well. Maybe something for the SI?

This is a great point. We now included the load plots in the SI for the other measured metals including Fe, Pb, Cd, Cu, Al, Mn, and Ni.

18. Figure 4: Please add the full legend.

We apologize for missing that. We added the full legend with the geochemical phases for Zn.

19. L407: For the combination of hysteresis and C-Q slopes, I recommend citing Winter et al. (2021) here as well.

Thanks for this great recommendation. Winter et al. (2021) provides a valuable framework for combining C-Q slopes with hysteresis analysis across multiple time scales, which is directly relevant to our study. We included this reference in the revised manuscript.

20. L409: A slope < -1 implies that not only concentrations but even the load (CxQ) decreases. Hence, it is not only a dilution of baseflow by freshly incoming water, but also the overall mass of metals declines, compared to what we had seen before the event. This is interesting. I could imagine it might be explained by the previous accumulation of metals during low flow that is not flushed out of the system, but I would appreciate the thoughts of the authors on this in the discussion.

21. At the same time, I would be careful to relate a slope of -0.55 to transport-limited behavior, as this is still a

22. L412: As I understand, solute concentrations do not rise, but decline. The entire paragraph needs to be carefully checked in line with my comment above.

23. L449: The peak alignment is interesting and might tell us something about the underlying processes. From my interpretation, it appears to emphasize the importance of hydrological processes, specifically dilution, but it does not align well with the theory of distant versus proximal sources.

Thank you for these very thoughtful and connected comments on the interpretation of dilution slopes, load dynamics, and peak alignments. We acknowledge that in the original draft, some of these aspects may have been simplified too much or not fully elaborated, which has caused some inconsistencies. We highly appreciate that you point us to these issues. Upon revisiting our data, we recognize that the strongest patterns at sites 3A and 3B reflect not only dilution (slope < −1) but also a decline in total metal loads (as seen in Fig. 4), pointing to depletion of previously accumulated solute pools. At site 2, by contrast, the slope of −0.55 reflects a weaker dilution signal, consistent with partial mobilization of solutes rather than purely transport-limited nor source-limited behavior. We further agree that the alignment of hysteresis peaks emphasizes hydrological processes of dilution such that whether there is a slower or faster reaction of concentration than discharge (but may be influenced by the hysteresis method used). Therefore, this does not necessarily map onto a simple proximal versus distal source interpretation, as you have pointed out. In the revised manuscript, we carefully revised Section 3.3

Institut für Mineralogie,· Brennhausgasse 14,· 09599 Freiberg · www.tu-freiberg.de/fakult3/

and associated figures to explicitly separate weak vs. strong dilution patterns, clarify that slopes < –1 may potentially indicate load depletion, reframe site 2 as partial mobilization rather than fully transport limitation, and present both dilution-driven and threshold-driven interpretations of peak alignment.

24. L515-546: Overall, I find this very convincing. However, I miss a critical discussion of what might be the drawbacks or, instead, the advantages of end-of-the-pipe solutions. I am not an expert, but I could imagine end-of-the-pipe solutions are easier to implement as they only need to be implemented once, and there might be less danger of missing specific spots? When only looking at specific hot spots, there might be a risk of overlooking other hot spots, especially in hydrologically complex mining-impacted systems, right?

We appreciate this excellent point. We agree that a critical discussion of end-of-pipe versus hotspot-focused remediation approaches strengthens the manuscript. In the revision, we framed end-of-pipe strategies not only in terms of their ease of implementation, but also as a potential solution that has so far received limited evaluation in the context of legacy mine systems. At the same time, hotspot-focused interventions offer the ability to intercept highly concentrated pulses but may miss other active sources in hydrologically complex settings. We further emphasized that the very dynamic nature of contaminant mobilization, which is rarely integrated into current monitoring or planning, suggests that the most effective approach may be a combination of system-scale safeguards with targeted diagnostics.

25. L540: Is that load in terms of PLI, or load in terms of C*Q?

The way load is used here is in terms of C*Q. We clarified this by stating "load from flux-based metrics".

**Referee #2**: This study by Sanchez et al explores the mechanisms of metal mobilisation at an abandoned mine site in Germany. The study uses high spatial and temporal resolution sampling to understand the complex subsurface hydrological processes and connectivity that drive metal release to surface waters.

In general, I found the paper to be an important scientific contribution, in particular in challenging the long-held assumption that wet periods and high river flows are always the most important periods for metal mobilisation and transport from abandoned mine sites. The paper presents high-quality analyses and methods that can help locate critical metal source areas in hydrologically [and geochemically] complex mine systems that can be used to inform remediation interventions. I found the spatial and temporal detail of the analyses to be excellent, and the implications and conclusions section present a very clear message that low spatial resolution and infrequent monitoring may lead to ineffective mine site characterisation and intervention measures.
My main critique concerns the detail and length of the introduction and methods sections. They are far too long and read more like a textbook chapter, which I feel confuses and dilutes important messages. I recommend Major Revisions to address these key areas.

Thank you very much for your thoughtful evaluation of our manuscript. We see that we have included a very lengthy introduction and methods section which may have diluted our important messages. This was now addressed along with the lines presented below.

**In text comments:**

Title:
Use of the words "hotspots" and "hot moments" in the title must be fully supported in the text by explanation of what these terms mean.

We fully agree that "hotspots" and "hot moments" must be explained in the main text given that they are in the title. We now made sure to be explicit with what these terms mean in lines 98-100.

Abstract:
L25. I presume the word "levels" here refers to concentrations? Please change.

This is an important point. By "levels" we refer to load amounts, not simply concentrations but concentration times discharge values. We changed this to the correct terminology.

Introduction:
This is an interesting and comprehensive evaluation of current literature and scientific understanding, but it is far too long in my opinion. I believe this could be made much shorter by focusing on the key unknowns and removal of complex explanations of, e.g. hysteresis patterns. Much of the more detailed discussion here could supplement interpretations later in the study. I think the complexity of the introduction is exemplified by the use of figure 1 to explain C-Q patterns and

Institut für Mineralogie, · Brennhausgasse 14, · 09599 Freiberg · www.tu-freiberg.de/fakult3/

hysteresis. It may be personal opinion, but I would not have figures in introductions, and I very rarely see it. The introduction should clearly and succinctly summarise the key challenges and unknowns.

We appreciate this thoughtful feedback. We agree that the introduction should be focused more sharply on the key unknowns and overarching research rationale. We reduced passages of blended focus. We checked carefully to streamline the arguments to focus the respective knowledge gaps without too much repetition. While we think that there is no single and crisp foundation for the C-Q patterns which just needs to be cited, we have moved the more detailed explanations in the methods section. We also acknowledged the point regarding the inclusion of a figure in the introduction. Since this goes along with the C-Q analysis explanations, we moved this figure and detailed explanations and revised this aspect in the Section 2.6.4: Conceptualization of site-specific C-Q patterns in the methods where it can better serve as a conceptual aid.

L121. The terms hot moments and hotspots are introduced without any definition of what they are. I don't believe these terms are necessary here.

This is a good point. We included these terms here to exemplify that we are building on our previous study to focus more in on these hotspots and hot moments at specific sites in the mine. We believe these terms are necessary to show that we do this by developing and answering our research questions and objectives. Therefore, we now explain what these terms mean.

The aims and objectives are far too long and complex here. You have hypotheses, research questions and objectives. Could you please revise to state a simple research question or objective and two or three related objectives?

Thanks for this helpful feedback. We realize that this is a bit too much. We agree that the aims and objectives can be simplified to improve clarity and focus. We streamlined this part by combining and condensing our two research questions into one overarching research question and by making our three objectives more concise and not overly complicated in lines 100-117.

Methods:
L141. Could you specific mineralogy here? What minerals are present? This is relevant for understanding metal mobilisation and attenuation processes.

Thank you for this suggestion. We agree that adding information on the dominant mineralogy helps to clarify the geochemical context. The host rock assemblage comprises primarily of gneiss and mica schist intersected by polymetallic sulfide-quartz-carbonate veins containing mainly pyrite, sphalerite, galena, and chalcopyrite. Since the sulfide-rich assemblages found at Reiche Zeche are key sources of acid generation and metal mobilization, this information is definitely relevant for understanding metal mobilization and attenuation processes. We added a concise description of this in lines 129-132.

L169. Could you clarify what is meant by "tracked" here?

Thanks for pointing this out. By "tracked" we mean measured. In the text, we changed this to say that we measured discharge, isotopic composition, and dissolved metal(loid) concentrations.

L170-189. There are quite a lot of preliminary data presented here. I understand the need to set the scene, but this could be condensed, and any discussion aspects removed.

Yes, this is a fair point. We do refer to our recent study and another study when trying to set the stage for our sampling design. The discussion about this may not need to be as detailed so we moved this to our results and discussion in subsection 3.1. We condensed this paragraph to focus on the most relevant background information that is suitable for this sampling design and conceptualization subsection in our methods.

L253. What is meant by "PDSI values"?

Thank you for highlighting this. We are using the Palmer Drought Severity Index (PDSI) as our water availability index. The PDSI quantifies surface moisture anomalies based on precipitation, evapotranspiration, and a simplified soil water balance model. It accounts for both short-term fluctuations and long-term storage effects, with its self-calibrating structure allowing the effective storage capacity to adjust dynamically to the amplitude of the local climate variability. The magnitude of PDSI indicates the severity of the departure from normal conditions, with values greater than +1 representing wet conditions and values below -1 indicating dry conditions. We clarified this in the text and adhere to the term "water availability index" for easier understanding.

L255. I presume "trying" should by "drying" here?

Thanks for catching this mistake. We fixed this to be "drying".

Institut für Mineralogie,· Brennhausgasse 14,· 09599 Freiberg · www.tu-freiberg.de/fakult3/

Overall, I feel the methods section is too long and could be condensed. Perhaps it is long because the methods are over-justified with a lot of explanation and over-referencing.

Thanks for this suggestion. We agree that the methods section is lengthy and has a lot of explanation. However, much of this information is needed to ensure that our analyses are well understood. We went through the methods section paragraph by paragraph and condensed the sections that may be overly explained or over-referenced.

Results and discussion:
L342. Please clarify what is meant by "top-most" site.

You're right. This needs clarification. By "top-most" site we meant the site that is located closest to the surface in our underground mine system. This site would be site 1 that is on level 1 (103 m below surface) in our study layout.

Generally, I think this section is presented very well. Insightful analyses and interpretations. I particularly like the implications section.

Conclusions: Short but impactful. Very clear take away message.

We are grateful for your nice feedback.

Thank you very much again to the editor and both referees for your thoughtful evaluation of the manuscript and your very constructive suggestions. Your comments helped us a lot to improve the paper towards coherence and clarity.

Sincerely,

Anita Alexandra Sanchez and Conrad Jackisch
(on behalf of all co-authors)

Institut für Mineralogie,· Brennhausgasse 14,· 09599 Freiberg · www.tu-freiberg.de/fakult3/

---

## Author Response (AR2)

**Institut für Bohrtechnik und Fluidbergbau**
Professur für Strömungs- und Transportmodellierung in
der Geosphäre

[Figure]

To the Editor and Referees of
Hydrological and Earth System Sciences

| Bearbeiter: | **Jun. Prof. Dr. Conrad Jackisch** |
|---|---|
| Anschrift: | Werner-Arnold-Bau, Agricolastraße 22, |
| Telefon: | 09599 Freiberg |
| E-Mail: | 03731 39-2681 |
| Homepage: | Conrad.Jackisch@tbt.tu-freiberg.de |
| | tu-freiberg.de/fakult3/ibf |
| Datum: | 06.12.2025 |

**Response to editor and referee**

We thank the editor and referee for their thoughtful evaluation of our revised manuscript "Hotspots and hot moments of metal mobilization: dynamic connectivity in legacy mine waters" and for accepting our manuscript, subject to implementation of the minor revisions suggested by the referee. We are very grateful for the important clarifications the referees requested for, which we addressed when revising the manuscript.

**Referee #1**: I have reviewed the manuscript titled "Hotspots and Hot Moments of Metal Mobilization: Dynamic Connectivity in Legacy Mine Waters" by Sanchez et al. for the second time. The authors have conducted a thorough revision, improving readability and avoiding ambiguity in their terminology. They have also corrected their C-Q and hysteresis analysis to provide a more precise conceptual representation of potential export patterns (see Fig. 2). The manuscript provides valuable insights into the complex water quality dynamics at abandoned mining sites, breaking down this complexity into dominant processes and hot moments. I have no doubt that this work will be of great value to readers of HESS. I have a few very minor comments, which I will leave to the authors' discretion as to whether they wish to incorporate them, and I recommend publication thereafter.

Thank you very much indeed for your thoughtful evaluation of our revised manuscript. We see that there are minor technical revisions that should be incorporated. We have now addressed the comments in the lines presented below.

L368: 'Observed' might be a better term than 'exhibited' here?

Thank you for highlighting this. We see how "observed" may be a better term in the context of the sentence. We have now replaced the original term.

L392-396: For better readability, consider splitting this sentence into two.

Thanks for noting this. We agree that the sentence is very long and could be better written as two sentences. We have now amended this to improve clarity and readability.

Fig. 3, 4 and 5: I cannot see any red indicating 'recession' in these figures. Therefore, it could be removed from the legend.

We're thankful for this observation. We agree that while we identify a recession phase in our geochemical classification framework, we do not observe this phase when developing our figures 3-5. We have now removed "recession" from the legend and stated in the figure caption that recession is not included because it was not identified here. We hope this improves the clarity of the figures.

L524: One might add here that C-Q slopes have been shown to diverge between the event and long-term scales (Winter et al., 2024; https://doi.org/10.1029/2024GL108437). This demonstrates the importance of analyzing C-Q relationships on short time scales using high-frequency data, as this reveals processes that would otherwise remain hidden. The study presented is a good example of why it is important to observe event-scale dynamics using high-frequency data rather than just long-term patterns using low-frequency data, as explained further below (hence, this may also fit into the next chapter.)

Thanks for this suggestion. We agree that by integrating this point we can make our conceptual focus more explicit and our results and discussion section stronger. We have added this information in lines 522-524, as well as in the next section 3.4 in lines 572-573 where it also fits.

[Figure]

[Figure]

[Figure]

Thank you very much again to the editor and referee for your thoughtful evaluation of the revised manuscript and your very nice suggestions. Your comments helped us to improve the paper towards coherence and clarity.

Sincerely,

Anita Alexandra Sanchez and Conrad Jackisch
(on behalf of all co-authors)

Institut für Mineralogie,· Brennhausgasse 14,· 09599 Freiberg · www.tu-freiberg.de/fakult3/